# Involvement and possible role of transglutaminases 1 and 2 in mediating fibrotic signalling, collagen cross-linking and cell proliferation in neonatal rat ventricular fibroblasts

Doa'a G. F. Al-U'datt[1]*, Carole C. Tranchant[2], Belal Al-Husein[3], Roddy Hiram[4,5], Ahmed Al-Dwairi[1], Mohammad AlQudah[1,6], Othman Al-shboul[1], Saied Jaradat[7], Jenan Alqbelat[1], Ali Almajwal[8]

1 Department of Physiology and Biochemistry, Faculty of Medicine, Jordan University of Science and Technology, Irbid, Jordan, 2 School of Food Science, Nutrition and Family Studies, Faculty of Health Sciences and Community Services, Université de Moncton, New Brunswick, Canada, 3 Department of Clinical Pharmacy, Faculty of Pharmacy, Jordan University of Science and Technology, Irbid, Jordan, 4 Montreal Heart Institute, Université de Montréal, Montreal, Quebec, Canada, 5 Department of Medicine, Université de Montréal, Montreal, Quebec, Canada, 6 Physiology Department, Arabian Gulf University, Manama, Bahrain, 7 Princess Haya Biotechnology Center, Jordan University of Science and Technology, Irbid, Jordan, 8 Department of Community Health Sciences, College of Applied Medical Sciences, King Saud University, Riyadh, Saudi Arabia

* dgaludatt@just.edu.jo

## Abstract

Transglutaminase (TG) isoforms control diverse normal and pathophysiologic processes through their capacity to cross-link extracellular matrix (ECM) proteins. Their functional and signalling roles in cardiac fibrosis remain poorly understood, despite some evidence of TG2 involvement in abnormal ECM remodelling in heart diseases. In this study, we investigated the role of TG1 and TG2 in mediating fibrotic signalling, collagen cross-linking, and cell proliferation in healthy fibroblasts by siRNA-mediated knockdown. siRNA for TG1, TG2 or negative control was transfected into cultured neonatal rat ventricular fibroblasts and cardiomyocytes. mRNA expression of TGs and profibrotic, proliferation and apoptotic markers was assessed by qPCR. Cell proliferation and soluble and insoluble collagen were determined by ELISA and LC-MS/MS, respectively. TG1 and TG2 were both expressed in neonatal rat cardiomyocytes and fibroblasts before transfection. Other TGs were not detected before and after transfection. TG2 was predominantly expressed and more effectively silenced than TG1. Knocking down TG1 or TG2 significantly modified profibrotic markers mRNA expression in fibroblasts, decreasing connective tissue growth factor (CTGF) and increasing transforming growth factor-β1 compared to the negative siRNA control. Reduced expression of collagen 3A1 was found upon TG1 knockdown, while TG2 knockdown raised α-smooth muscle actin expression. TG2 knockdown further increased fibroblast proliferation and the expression of proliferation marker cyclin D1. Lower insoluble collagen content and collagen cross-linking were evidenced upon silencing TG1 or TG2. Transcript levels of collagen 1A1, fibronectin 1, matrix metalloproteinase-2, cyclin E2, and

**Data Availability Statement:** All relevant data are within the paper and its Supporting Information files.

**Funding:** This study was supported by Jordan University of Science and Technology/Deanship of Research/Project number (2020/166). The funders had no role in the study design, data collection, and analysis, decision to publish, or preparation of the manuscript.

**Competing interests:** The authors declare that they have no competing interests.

BCL-2-associated X protein/B-cell lymphoma 2 ratio were strongly correlated with TG1 mRNA expression, whereas TG2 expression correlated strongly with CTGF mRNA abundance. These findings support a functional and signalling role for TG1 and TG2 from fibroblasts in regulating key processes underlying myocardial ECM homeostasis and dysregulation, suggesting that these isoforms could be potential and promising targets for the development of cardiac fibrosis therapies.

## Introduction

Cardiac fibrosis, a ubiquitous pathophysiological process in heart diseases, is defined as an excessive deposition of extracellular matrix (ECM) proteins that eventually leads to increased myocardial stiffness and impaired cardiac functions [1]. Cardiac fibroblasts perform central roles in ECM remodelling and fibrosis, being responsible for proteins synthesis and degradation, structural support, cellular proliferation, inflammation, as well as scar formation following cardiac injury [2]. Recent evidence suggests that despite representing a relatively small fraction of the total cardiac cells [3], fibroblasts are involved in numerous cross-talks with other cardiac cells, including cardiomyocytes and endothelial cells. Collagens I and III are the most important fibrillar proteins secreted in the myocardial ECM under pathological conditions [4]. Excessive cross-linking of collagen fibers, whether by enzymatic or non-enzymatic mechanisms, is a major contributor to the progression of cardiac fibrosis. Lysyl oxidases (LOX), LOX-like isoforms, and transglutaminases (TGs) are the main collagen cross-linking enzymes [5–8].

TGs are calcium-dependent thiol enzymes that catalyze the formation of covalent isopeptide bonds between the free amine group of lysine residues and the γ-carboxamide group of glutamine residues. The resistance of TG cross-links to the action of proteases confers stability and rigidity to the ECM [7, 9]. In higher vertebrates, the TG family is comprised of nine isoforms, namely TG1 to TG7, blood coagulation factor XIII, and erythrocyte band 4.2 [9]. Keratinocyte TG (or TG1) has been detected in the skin epidermis, brain, and myocardial and vascular endothelial cells [10, 11]. Tissue TG (TG2), the most studied and ubiquitously expressed TG isoform, is found in diverse types of tissues and cells, including cardiac endothelial cells and fibroblasts, vascular smooth muscle cells, monocytes, and macrophages [12–14]. In addition to its transamidase (cross-linking) activity, this multifunctional protein regulates numerous cellular functions such as growth, differentiation, migration, adhesion, survival, apoptosis, and angiogenesis [12–15]. A growing body of evidence shows that TG2 is pivotal to the development of cardiac diseases such as myocardial hypertrophy, myocardial infraction and heart failure through its suspected involvement in cardiac fibrosis signalling processes [16]. Increased TG2 level was reported in a pressure-overloaded heart induced by transverse aortic constriction [16]. In myocardial infarction and pressure-overload models, TG2 has been identified as a promising therapeutic target against cardiac fibrosis [17].

TGs intra- and extracellular functions in cardiac physiology and disease have not yet been fully elucidated [18]. One of TG2 extracellular functions in normal ECM remodelling processes and cardiac fibrosis is collagen cross-linking due to its transamidase enzymatic activity [19–21]. Studies in animal models involving pressure overload, hypoxia or volume overload in different cardiac diseases (e.g., hypertrophic cardiomyopathy, myocardial ischemia, and heart failure) have showed that cardiac fibrosis is concomitant with increased TG2 expression and cross-linking of ECM proteins [17, 19, 21–24]. While TG2 was recently identified as the predominant isoform in heart tissue [17, 19, 21–24], little is known about it transamidase-

independent (signalling) actions in cardiac cells. Moreover, it is unclear what roles, if any, other isoforms may play in these cells. Some of the newly identified cellular signalling roles of TGs, including cell proliferation, apoptosis, and gene transcription [18, 21], have seldom been investigated in cardiac cells. The present study aimed to evaluate the roles of TG1 and TG2 in mediating fibrotic signalling, collagen cross-linking, as well as fibroblast proliferation in healthy fibroblasts by silencing RNA (siRNA)-mediated knockdown. mRNA expression of TG1, TG2, TG3, TG4, and TG6 was assessed to determine whether TG1 or TG2 knockdown may elicit compensatory expression of other isoforms.

## Materials and methods

### Animal model

All the animal handling procedures and care protocols were approved by the Animal Research Ethics Committee of Jordan University of Science and Technology and followed the guidelines from the Jordanian Animal Care Council. Male pups aged 2–3 days from Wistar rats were obtained from the University Animal House. Neonatal rat pups were separated from their mother and were pre-anesthetized by hypothermia (indirect contact with ice as the pups were wrapped in gaze) to lessen any stress on them. Their abdomen was sterilized with 70% ethanol for local anesthesia before they were euthanized by decapitation with sharp surgical scissors by a trained researcher. Their chest was cut open with sharp scissors and the beating heart was quickly excised, then transferred to HBSS medium (sterile calcium and magnesium-free Hank's Balanced Salt Solution) for isolation of fibroblasts and myocytes. Each series of cell isolation and culture experiments was performed with 13 to 15 pups. The cell isolation and culture process was repeated one to three times until enough cells were collected.

### Isolation of cardiomyocytes and fibroblasts from neonatal rat ventricles

Ventricular fibroblasts and cardiomyocytes were isolated from neonatal hearts according to the method described by Duong et al. [25]. All the reagents and enzymes used were purchased from Worthington Biochemical (Lakewood, NJ, USA) unless otherwise indicated. Briefly, the ventricles were separated from the atria, minced into small pieces, and partially digested with trypsin (50 μg/mL) at 4˚C for 16 h, followed by addition of soybean trypsin inhibitor (2 mg/mL). Next, the samples were mixed with 5 mL of collagenase (300:1 (v/v) of enzyme in Leibovitz L-15 medium) and incubated for 30 min at 37˚C with constant agitation before centrifugation for 5 min at $60 \times g$ to separate the cardiomyocytes. Pelleted cardiomyocytes were resuspended in Gibco Medium 199 (Themo Fisher Scientific, NY, USA) supplemented with fetal bovine serum (FBS, 10%), 0.2% insulin-transferrin-selenium, and 1% penicillin-streptomycin. Cells were purified into a highly pure population using a 40–60% Percoll step gradient (GE Healthcare Life Sciences, Brondby, Denmark) as described by Golden et al. [26]. Fibroblasts were separated by sequential centrifugation of the supernatant at increasing speed (5 min at $60 \times g$ and 5 min at $300 \times g$).

### Culture and siRNA transfection of ventricular cardiomyocytes and fibroblasts

Isolated ventricular cardiomyocytes and fibroblasts were counted with a hemocytometer and cultured in 6-well plates at 500,000 cells/well. The plates were incubated at 37˚C for 24 h and the cells were subsequently washed with growth medium (10% FBS in Medium 199). After 48 h, the medium was changed to serum-deprived medium (0% FBS) for 24 h, then the cardiomyocytes and fibroblasts were transfected with small interfering RNA (siRNA) for TG1

(siTG1), TG2 (siTG2) or negative silencer control (NegsiRNA) (assay IDs 133272, 132701 and 4390843, respectively; Invitrogen, Thermo Fisher Scientific, NY, USA). SiRNA transfection was performed according to the manufacturer's instruction by incubating the cells with siRNA (100 nM) in Opti-MEM medium (Gibco) containing Lipofectamine RNAiMAX (Invitrogen, Thermo Fisher Scientific). Cells and supernatants were harvested 24 h after transfection for further analyses.

## Cell proliferation assay

Fibroblast proliferation was expressed as the number of viable cells determined with a colorimetric enzyme-linked immunosorbent assay (ELISA) (Cell Counting Kit-8, Dojindo, Kumamoto, Japan), as previously described Li et al. [27]. This assay relies on the biological reduction of a water-soluble tetrazolium salt (WST-8) by the dehydrogenase enzymes present in viable cells, which produces a yellow-colored formazan dye. Ventricular fibroblasts were plated onto 96-well plates (2000 cells/100 μL in each well) and cultured in complete growth medium (10% FBS in Medium 199). After 24 h, the medium was changed to FBS-free medium, followed by addition of 10 μL of WST-8 solution per well, mixing and incubation at 37˚C for 2 h. The absorbance was read at 450 nm in a microplate reader (SpectraMax 190, Molecular Devices, San Jose, CA, USA). The results were expressed as relative to the negative control (NegsiRNA).

## mRNA expression by quantitative real-time polymerase chain reaction (RT-qPCR)

mRNA gene expression of TG isoforms (TG1, TG2, TG3, TG4, TG6), profibrotic markers [collagen 1A1 (COL 1A1), collagen 3A1 (COL 3A1), fibronectin 1 (FN 1), α-smooth muscle actin (α-SMA), connective tissue growth factor (CTGF), transforming growth factor β1 (TGF-β1), matrix metalloproteinases 2 and 9 (MMP-2 and MMP-9), periostin and connexin 43 (CX 43)], proliferation markers [cyclins D1 and E2 (CCND 1 and CCNE 2)], and apoptotic markers [B-cell lymphoma 2 (BCL-2) and BCL-2-associated X protein (BAX)] were determined as described by Duong et al. [25]. Transfected cardiomyocytes and fibroblasts were collected in TRIzol reagent (Zymo Research, Irvine, CA, USA) to extract the total RNA, which was purified using a Direct-zol RNA MiniPrep kit (Zymo Research, USA). Quantification and qualification of RNA were assessed using a NanoDrop™ 2000 spectrophotometer (Thermo Fisher Scientific, Wilmington, USA).

Complementary DNA (cDNA) was synthesized from 250 ng of total RNA using a High Capacity cDNA Reverse Transcription Kit supplied by Applied Biosystems (Foster City, CA, USA), USA). RT-qPCR was performed with a PreAmp Master Mix (Takara Bio Inc., Shiga, Japan) and TaqMan Universal PCR Master Mix (Applied Biosystems). Custom-designed primers for α-SMA, CTGF, TGF-β1, MMP-2, MMP-9, periostin, CX 43, CCND 1, CCNE 2, BAX, BCL-2, and β2 microglobulin (S1 Table) were purchased from Integrated DNA Technologies (Coralville, IA, USA). TaqMan probes for TG1 (assay ID Rn00581408_m1), TG2 (Rn00571440_m1), TG3 (Rn01513750_m1), TG4 (Rn00575599_m1), TG6 (Rn01747645_m1), COL 1A1 (Rn01463848_m1), COL 3A1 (Rn01437681_m1), and FN 1 (Rn00569575_m1) were supplied by Applied Biosystems (Foster City, CA, USA). mRNA expression levels were normalized to the internal standard β2 microglobulin according to the proportional threshold cycle ($2^{-\Delta\Delta Ct}$) method proposed by Livak and Schmittgen [28]. The obtained values were expressed as relative to the value obtained for the control (NegSiRNA) to show the change of gene expression relative to the control.

## Determination of collagen content

The concentration of hydroxyproline in the hydrolyzed fibroblast cells and supernatants was used to estimate the amount of insoluble collagen and soluble collagen, respectively. The ratio of insoluble to soluble collagen, an indication of the amount and possibly the nature of cross-links present in collagen molecules [29], was also calculated. Cells and supernatants were dried in a centrifugal vacuum concentrator (Labconco CentriVap, Labconco, Kansas City, MO, USA) before hydrolysis. The dried cells and supernatants were dissolved in 200 μL of 6 M HCl and 12 M HCl, respectively, and hydrolyzed at 95˚C for 20 h using a Bio-Rad S1000 Thermal Cycler (BioRad, Mississauga, ON, Canada), followed by centrifugation to remove impurities and drying of the supernatants. In preparation for liquid chromatography-tandem mass spectrometry (LC-MS/MS), the dried samples were dissolved in 400 μL of 50% methanol in deionized water, then 50 μL of each aliquot was diluted in 1 mL of 50% methanol. The final aliquots were used for hydroxyproline determination by LC-MS/MS. All the solvents were of HPLC grade (Thermo Fisher Scientific).

Hydroxyproline content was quantified using a gradient LC-MS/MS electrospray ionization (ESI) system consisting of an Agilent HPLC module (1200 Series, Agilent Technologies, Stuttgart, Germany) coupled with an API 3200 mass spectrometer (AB SCIEX, Concord, ON, Canada). Hydrolyzed samples or hydroxyproline standard (10 μL each) were injected in triplicate into the system. Chromatographic separation was achieved at 45˚C on a Luna Omega PS C18 column UHPLC (50 × 2.1 mm; particle size 1.6 μm; Phenomenex Inc., CO, USA) with a mobile phase consisting of solvents A (0.1% formic acid in deionized water) and B (0.1% formic acid in absolute methanol) eluted at 0.7 mL/min. The separation gradient started with 2% of solvent B, increasing to 30% over 2 min, then to 90% for 30 s before reestablishing the initial composition. Conditions at the ESI-MS interface were 100˚C for source temperature, 50L/h for nitrogen desolvation gas flow, and 50 L/h for nitrogen cone gas flow. Ionization in the positive mode was performed at a spray voltage of 5.5 kV. AB SCIEX Analyst software v. 1.6.3 was used for data acquisition and processing. A stock solution of 1 000 ng/mL of hydroxyproline in solvent A was serially diluted in this solvent to prepare a standard curve from which the collagen content was determined and expressed relatively to the control.

## Statistical analyses

Data are reported as means of three or four replicates ± standard error of the mean (SEM) unless stated otherwise. Unpaired Student's *t*-tests were carried out for comparison between two groups. One-way analysis of variance (ANOVA) followed by Dunnett's multiple comparison test was performed for multiple comparisons after checking for normality and homogeneity of variances. Linear regression analysis was performed to assess the associations between TG1 or TG2 mRNA levels and the transcript levels of the individual profibrotic, proliferation and apoptotic markers. Statistical analyses were conducted with GraphPad Prism 8 (GraphPad Software, La Jolla, CA, USA) with statistical significance set at $P \leq 0.05$.

## Results

### TG1 and TG2 expressions were suppressed in neonatal rat ventricular fibroblasts and cardiomyocytes by individual siRNA

As illustrated in Fig 1, TG1 and TG2 mRNA expressions were efficiently silenced through siRNA-mediated knockdown in cultured neonatal rat ventricular fibroblasts and cardiomyocytes. Knocking down TG1 significantly reduced TG1 and TG2 mRNA expression in fibroblasts (Fig 1A, $P < 0.05$; Fig 1B, $P < 0.001$, respectively) and cardiomyocytes (Fig 1C,

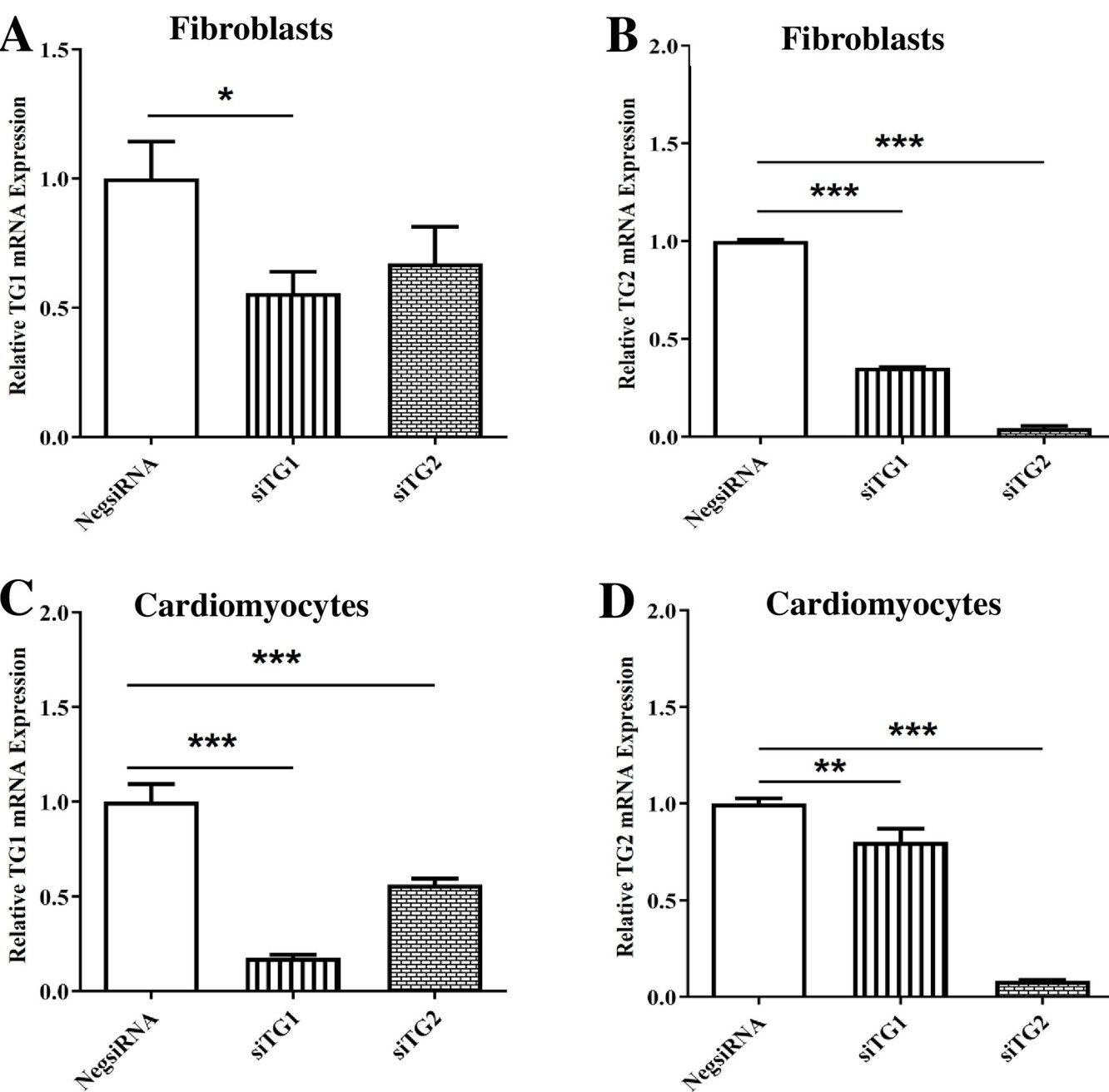

**Fig 1. Transglutaminase 1 (TG1) and TG2 expressions were efficiently suppressed in neonatal rat ventricular fibroblasts and cardiomyocytes using individual small interfering RNA (siRNA) for downregulating TG isoforms.** The specificity and efficiency of TG1 and TG2 knockdown were estimated by qPCR following transfection of neonatal rat ventricular fibroblasts and cardiomyocytes (n = 4 replicates). Effect of transfection with siTG1 or siTG2 on the relative TG1 and TG2 mRNA levels in fibroblasts (A and B) and cardiomyocytes (C and D) as compared with the negative control (NegsiRNA). The abundance of TG1 and TG2 mRNA was normalized to that of β2 microglobulin. Results are means ± SEM; unpaired Student's t-tests were performed. $^*P < 0.05$; $^{**}P < 0.01$; $^{***}P < 0.001$ vs. NegsiRNA.

$P < 0.001$; Fig 1D, $P < 0.01$). In contrast, TG2 silencing significantly suppressed TG2 expression in fibroblasts (Fig 1B, $P < 0.001$) with no significant change in TG1 expression (Fig 1A), while in cardiomyocytes TG2 knockdown reduced the expression of both isoforms (Fig 1C, $P < 0.001$; Fig 1D, $P < 0.001$). As shown in Fig 1B and 1D, TG2 was more effectively silenced

by its corresponding siRNA than TG1. Overall, these results indicate greater specificity of the siRNA for TG2 in fibroblasts as it effectively depressed TG2 transcription without affecting that of TG1. Of note, the other isoforms tested in this study, namely TG3, TG4, and TG6, were not detected in either fibroblasts or cardiomyocytes before and after transfection.

## Knockdown of TG1 and TG2 diminished insoluble collagen content and collagen cross-linking in neonatal rat ventricular fibroblasts

Knocking down TG1 or TG2 did not significantly affect the amount of soluble collagen in cultured fibroblasts as compared to the control (Fig 2A) but significantly decreased by more than half the insoluble collagen content (0.24±0.01 and 0.31±0.003-fold change, respectively; Fig 2B) and collagen cross-linking ratio (0.26±0.02 and 0.34±0.02-fold change, respectively; Fig 2C). These findings strongly suggest a functional role for TG1 and TG2 for collagen cross-linking in ventricular fibroblasts.

## Knockdown of TG1 and TG2 modified the mRNA expression of several profibrotic markers in neonatal rat ventricular fibroblasts

Knocking down TG1 or TG2 significantly reduced CTGF mRNA abundance compared to the control (0.46±0.10 and 0.22±0.07-fold change, respectively; Fig 3D), while upregulating TGF-β1 transcription (1.68±0.20 and 1.63±0.09-fold change; Fig 3H). COL 3A1 expression was reduced upon TG1 knockdown (0.84±0.01-fold change; Fig 3A), whereas TG2 silencing increased the transcription of α-SMA (1.23±0.02-fold change; Fig 3G). Transcription of the other profibrotic markers, namely COL 1A1, FN 1, MMP-2, MMP-9, CX 43, and periostin, remained unchanged upon TG1 or TG2 knockdown (Fig 3B, 3C, 3E, 3F, 3I and 3J). These findings suggest a role for TG1 and TG2 in regulating the expression of CTGF, TGF-β1, as well as COL 3A1 and α-SMA in ventricular fibroblasts.

## Knockdown of TG2 increased cell proliferation and mRNA expression of CCND 1 in neonatal rat ventricular fibroblasts

Knocking TG2 significantly increased fibroblast proliferation (1.47±0.10-fold change) as well as CCND 1 mRNA abundance (1.12±0.03-fold change) in fibroblasts, as shown in Fig 4A and 4B, respectively. mRNA expression of the other proliferation marker (CCNE 2) and of the apoptotic markers BAX, BCL-2, and BAX/BCL-2 ratio remained unchanged upon silencing TG1 or TG2 (Fig 4C–4F). These findings point to a possible implication of TG2 in regulating the proliferative capacity of ventricular fibroblasts.

## TG1 mRNA expression was strongly correlated with mRNA levels of COL 1A1, FN 1, MMP-2, CCNE 2, and BAX/BCL-2 ratio

As summarized in Fig 5 and Table 1, TG1 mRNA expression was strongly and positively associated with the transcript levels of COL 1A1, FN 1 and MMP-2 and BAX/BLC-2 ratio ($R^2$ = 0.666, 0.753, 0.863, and 0.819, respectively, $P < 0.001$; Fig 5B, 5C, 5E, and 5M), while a strong and negative correlation was found with proliferation marker CCNE 2 ($R^2$ = 0.547, $P < 0.001$; Fig 5O). Weaker negative correlations were evidenced with TGF-β1 and BCL-2 ($R^2$ = 0.272 and 0.274, $P$ = 0.03; Fig 5H and 5L). No significant correlations were found in the other profibrotic, proliferation and apoptotic markers investigated in this study. These findings suggest a possible association between TG1 transcription levels and the expression of certain profibrotic, proliferation and apoptotic markers in ventricular fibroblasts.

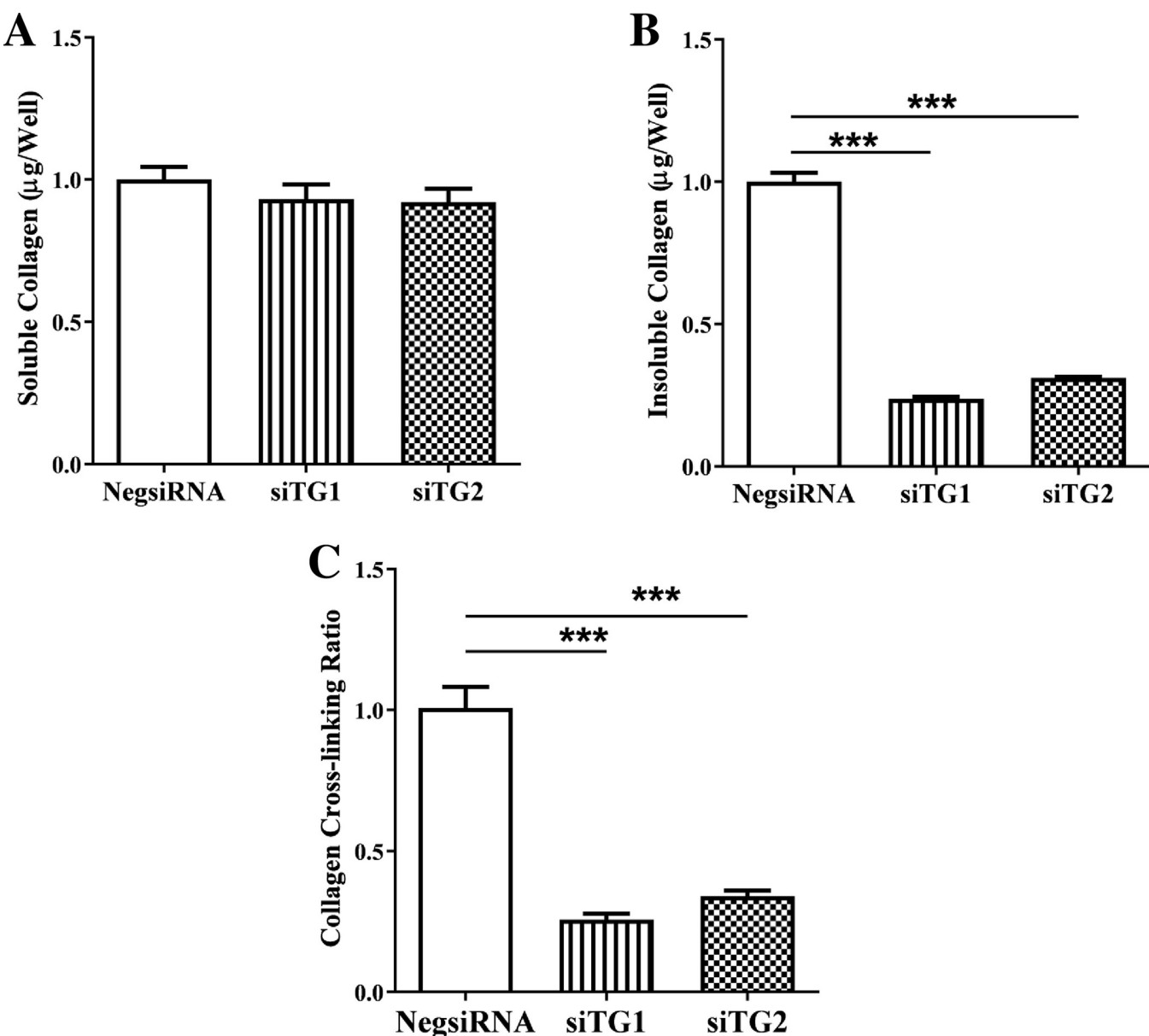

**Fig 2. Knockdown of transglutaminase 1 (TG1) and TG2 isoforms diminished the insoluble collagen content and collagen cross-linking ratio without detectable change in soluble collagen in neonatal rat ventricular fibroblasts.** Effect of transfection with siTG1 or siTG2 in fibroblasts (n = 3 replicates) on relative (A) soluble collagen content, (B) insoluble collagen content, and (C) collagen crosslinking ratio as compared with the negative control (NegsiRNA). Results are means ± SEM; one-way ANOVA followed by Dunnett's multiple comparison test was performed. ***$P < 0.001$ vs. NegsiRNA.

## TG2 mRNA expression was strongly correlated with CTGF mRNA expression

TG2 mRNA expression was strongly and positively correlated with CTGF mRNA level ($R^2 = 0.616$, $P < 0.001$; Fig 6D, Table 1). Significant, albeit weaker, positive correlations were found with mRNA abundance of TGF-β1, MMP-9, periostin, CCND1, BAX, and BCL-2 ($R^2 = 0.231$, 0.219, 0.233, 0.249, 0.325, and 0.282, $P = 0.04$, 0.05, 0.04, 0.04, 0.01, and 0.02, respectively; Fig 6, Table 1). No other significant correlations were found. These data indicate a possible

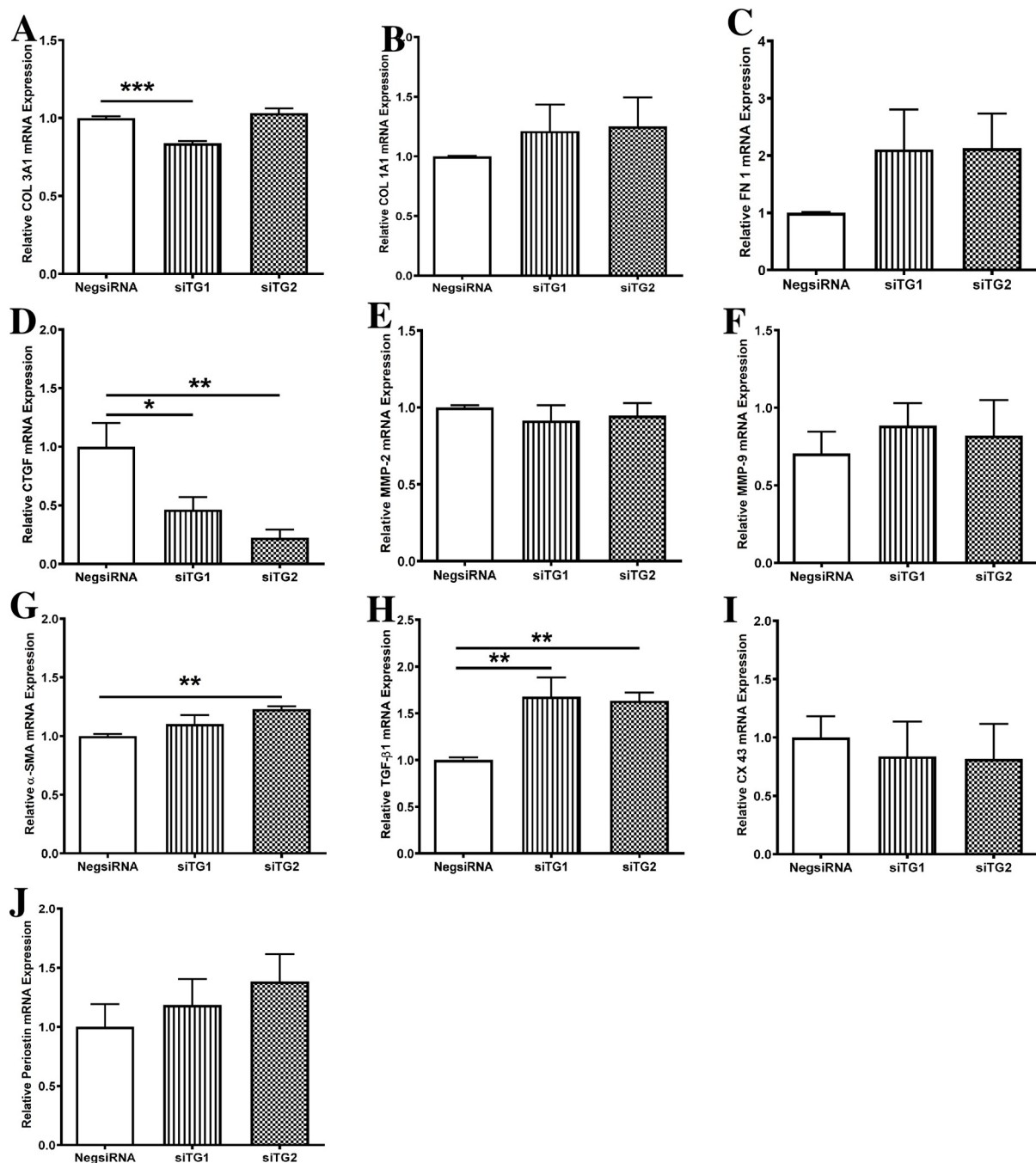

**Fig 3. Knockdown of transglutaminase 1 (TG1) and TG2 modified the mRNA expression of several profibrotic markers in neonatal rat ventricular fibroblasts.** Effect of transfection of ventricular fibroblasts with siTG1 or siTG2 (n = 4 replicates) on the relative mRNA expression of (A) collagen 3A1 (COL3 A1), (B) COL 1A1, (C) fibronectin 1 (FN 1), (D) connective tissue growth factor (CTGF), (E) matrix metalloproteinase-2 (MMP-2), (F) MMP-9, (G) α-smooth muscle actin (α-SMA), (H) transforming growth factor β1 (TGF-β1), (I) Connexin 43 (CX 43), and (J) periostin. mRNA abundance of profibrotic markers was normalized to that of β2 microglobulin. Results are means ± SEM; one-way ANOVA followed by Dunnett's multiple comparison test was performed. $^*P < 0.05$ vs. Negative control (NegsiRNA); $^{**}P < 0.01$ vs. NegsiRNA; $^{***}P < 0.001$ vs. NegsiRNA.

association between TG2 transcription levels and the expression of certain profibrotic, proliferation and apoptotic markers in ventricular fibroblasts.

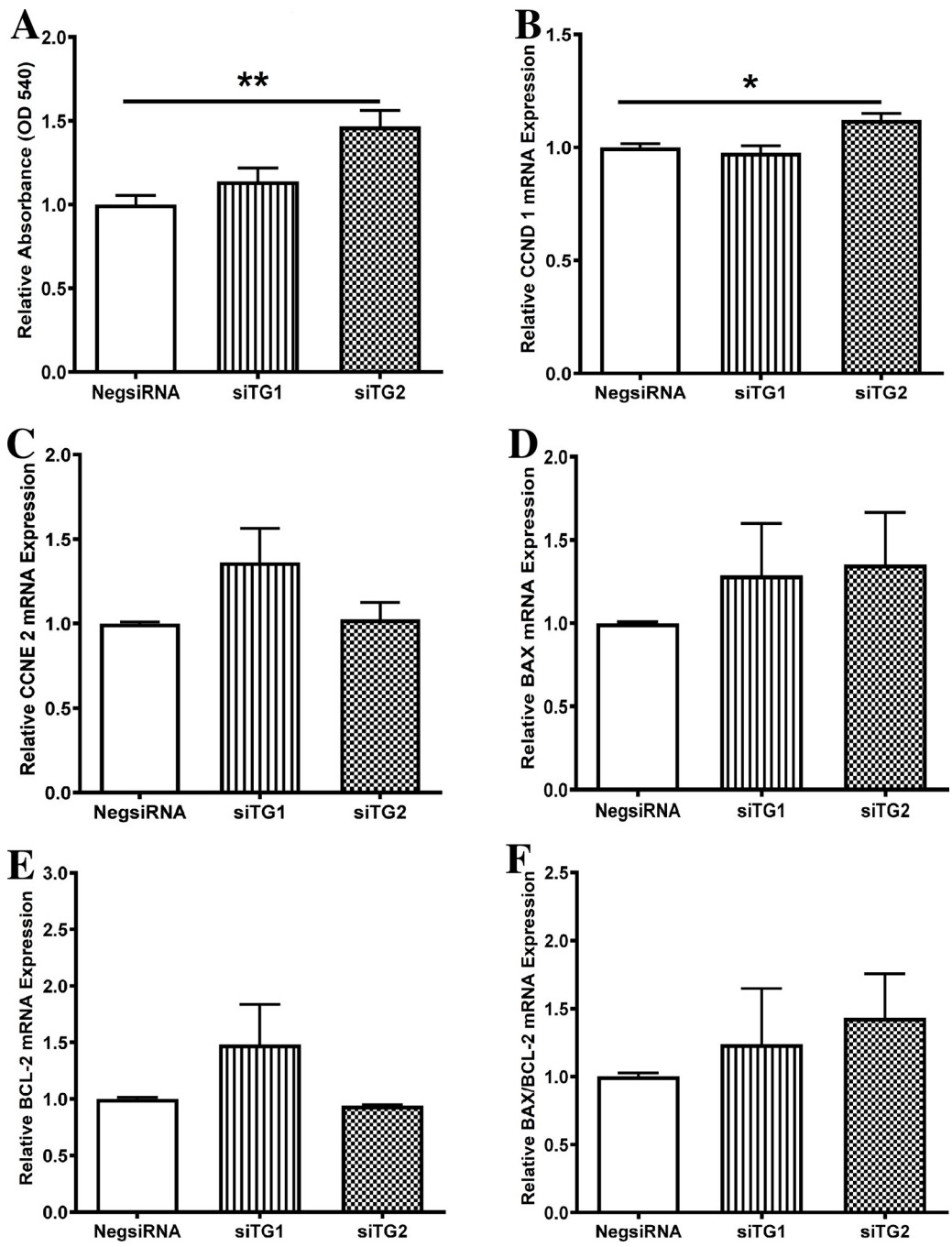

**Fig 4. Knockdown of transglutaminase 2 (TG2) in neonatal rat ventricular fibroblasts increased cell proliferation and mRNA expression of the CCND 1 proliferation marker.** Effect of transfection of ventricular fibroblasts with siTG1 or siTG2 (n = 4 replicates) on (A) relative cell viability (proliferation) and relative mRNA levels of (B) cyclin D1 (CCND 1), (C) cyclin E2 (CCNE 2), (D) B-cell lymphoma 2 (BCL-2)-associate X protein (BAX), and (E) BCL-2, and (F) BAX/ BCL-2 ratio. mRNA abundance of apoptotic and proliferation markers was normalized to that of β2 microglobulin. Results are means ± SEM; one-way ANOVA followed by Dunnett's multiple comparison test was performed. $^{*}P < 0.05$ vs. Negative control (NegsiRNA); $^{**}P < 0.01$ vs. NegsiRNA.

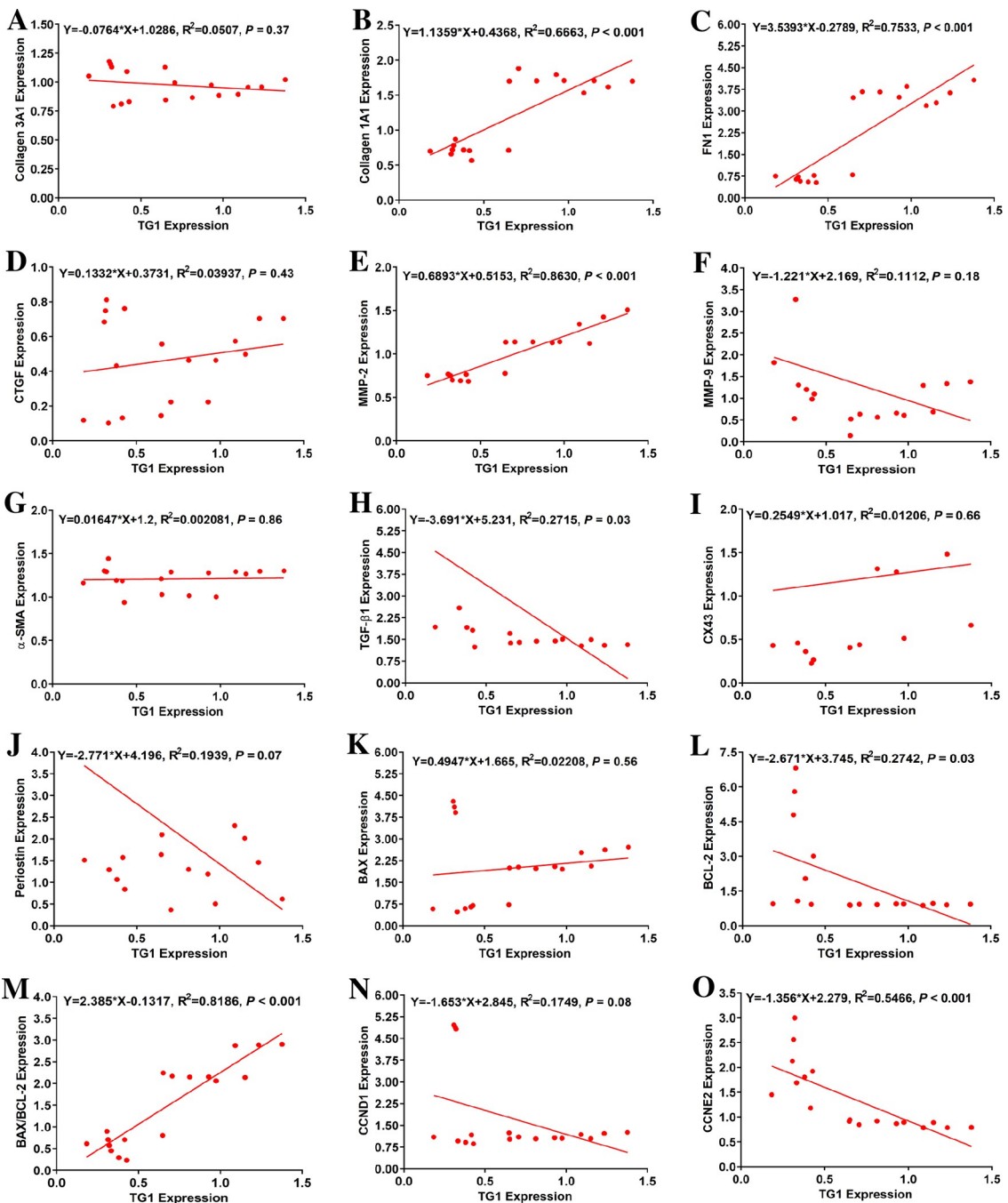

**Fig 5. Transglutaminase 1 (TG1) mRNA expression was strongly correlated with mRNA expression of collagen 1A1 (COL 1A1), fibronectin 1 (FN 1), matrix metalloproteinase-2 (MMP-2), cyclin E2 (CCNE 2) and BCL-2-associate X protein (BAX)/B-cell lymphoma 2 (BCL-2) ratio.** Correlations between mRNA levels of TG1 following individual knockdown and mRNA levels of profibrotic, proliferation and apoptotic markers in neonatal rat ventricular fibroblasts (n = 18): (A) collagen 3A1 (COL 3A1), (B) COL 1A1, (C) fibronectin 1 (FN 1), (D) connective tissue growth factor (CTGF), (E) matrix metalloproteinase-2 (MMP-2), (F) MMP-9, (G) α-smooth muscle actin (α-SMA), (H) transforming growth factor β1 (TGF-β1), (I) connexin 43 (CX 43), (J) periostin, (K) BAX, (L) BCL-2, (M) BAX/BCL-2 ratio, (N) cyclin D1 (CCND 1), and (O) cyclin E2 (CCNE 2). $R^2$: Correlation coefficient; in the equations, (X) and (Y) respectively denote TG1 expression and expression of the profibrotic, proliferation or apoptotic markers.

**Table 1. Correlation analysis between mRNA levels of TG1 or TG2 following individual knockdown and mRNA expression of profibrotic, proliferation, and apoptotic markers.**

| | TG1 | | TG2 | |
|---|---|---|---|---|
| Biomarker | $R^2$ | *P*-value | $R^2$ | *P*-value |
| **Profibrotic markers** | | | | |
| COL 1A1 | **0.666** | **<0.001** | 0.016 | 0.62 |
| COL 3A1 | 0.051 | 0.37 | 0.002 | 0.87 |
| FN 1 | **0.753** | **<0.001** | 0.003 | 0.84 |
| α-SMA | 0.002 | 0.86 | 0.004 | 0.81 |
| CTGF | 0.039 | 0.43 | **0.616** | **<0.001** |
| TGF-β1 | 0.272 ǂ | **0.03** | 0.231 | **0.04** |
| MMP-2 | **0.863** | **<0.001** | 0.015 | 0.63 |
| MMP-9 | 0.111 | 0.18 | 0.219 | **0.05** |
| Periostin | 0.194 | 0.07 | 0.233 | **0.04** |
| CX 43 | 0.012 | 0.66 | 0.199 | 0.06 |
| Proliferation markers | | | | |
| CCND 1 | 0.175 | 0.08 | 0.249 | **0.04** |
| CCNE 2 | **0.547 ǂ** | **<0.001** | 0.200 | 0.06 |
| Apoptotic markers | | | | |
| BAX | 0.022 | 0.56 | 0.325 | **0.01** |
| BCL-2 | 0.274 ǂ | **0.03** | 0.282 | **0.02** |
| BAX/BCL-2 ratio | **0.819** | **<0.001** | 0.004 | 0.82 |

*P*-values in bold indicate a statistically significant correlation. ǂ denotes negative correlations. Collagen 1A1: COL 1A1; Collagen 3A1: COL 3A1; Fibronectin 1: FN 1; α-smooth muscle actin: α-SMA; Connective tissue growth factor: CTGF; Transforming growth factor β1: TGF-β1; Matrix metalloproteinase-2: MMP-2; Matrix metalloproteinase-9: MMP-9; Connexin 43: CX43, Cyclin D1: CCND 1; Cyclin E2: CCNE 2; BCL-2-associated X protein: BAX; B-cell lymphoma 2: BCL-2.

## Discussion

This study provides novel evidence on the expression and implication of different TGs in ECM homeostasis and dysregulation in cardiac cells, as there are limited reports on the expression and roles of TG isoforms in cardiac fibroblasts and myocytes under physiological and pathological conditions. We began by showing that TG1 and TG2 were both transcriptionally expressed in neonatal rat ventricular fibroblasts and cardiomyocytes under nonpathological conditions, unlike TG3, TG4 and TG6. We then used *in vitro* siRNA transfection to assess the involvement of TG1 and TG2 in modulating collagen cross-linking, fibroblast proliferation and the expression of ECM profibrotic, proliferation, and apoptotic markers by fibroblasts. The main findings and novel contributions show that: (a) TG1 and TG2 transcription in fibroblasts and cardiomyocytes was efficiently suppressed by siRNA; (b) silencing TG1 or TG2 decreased the amount of insoluble collagen and collagen cross-linking in fibroblasts, together with altering the transcription levels of profibrotic markers CTGF, TGF-β1 (siTG1 or siTG2), COL 31A (siTG1), and α-SMA (siTG2); (c) TG2 knockdown increased fibroblast proliferation and transcription of proliferation marker CCND 1; and (d) TG1 expression was strongly correlated with the transcript levels of profibrotic (COL 1A1, FN 1, MMP-2), proliferation (CCNE 2), and apoptotic (BAX/BCL-2 ratio) markers, whereas TG2 expression correlated strongly with that of CTGF.

We found that the siRNA for TG2 specifically depressed the expression of this isoform in fibroblasts, whereas TG1 siRNA also partially depressed TG2 in addition to TG1. This is not

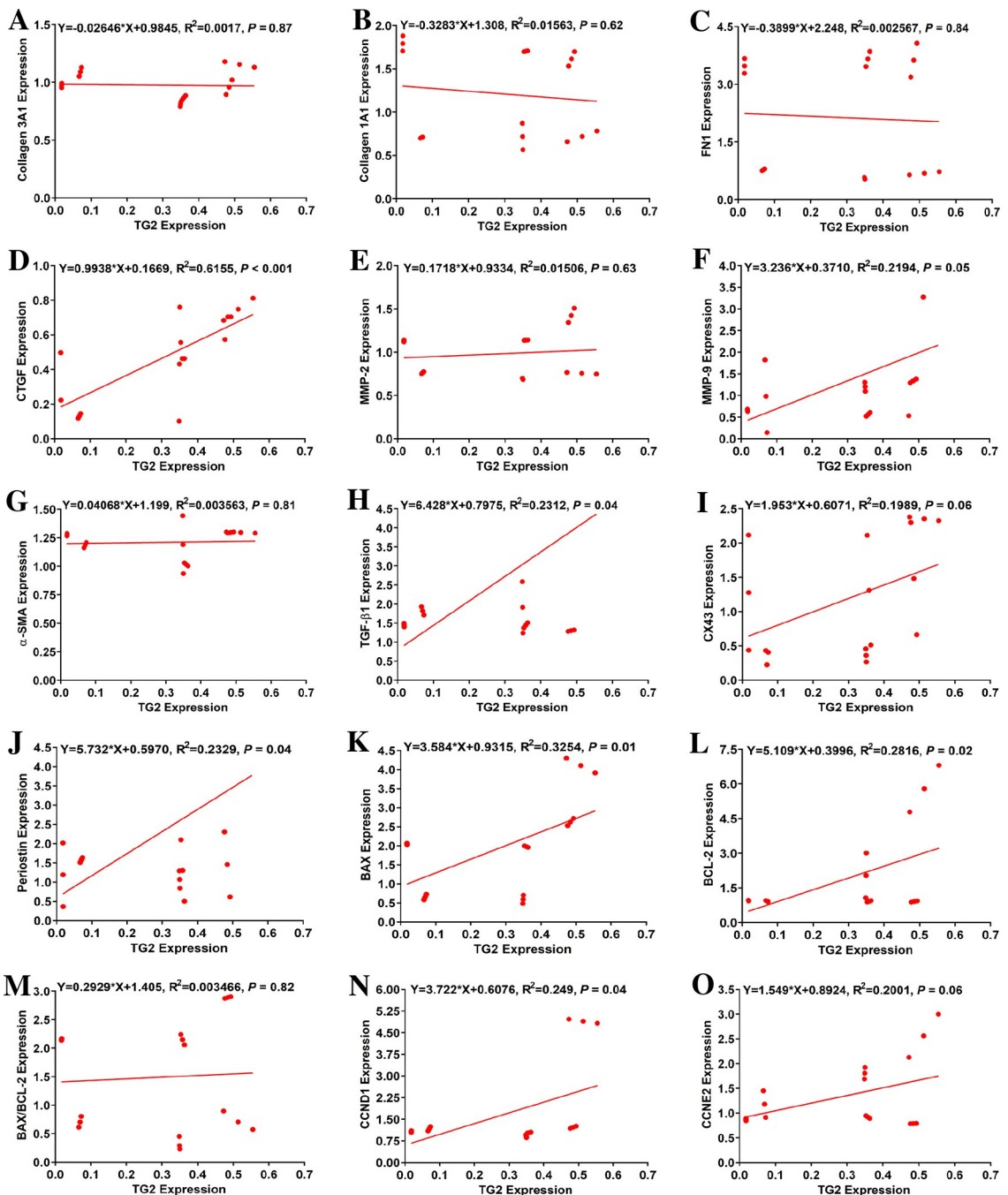

**Fig 6. Transglutaminase 2 (TG2) mRNA expression was strongly correlated with mRNA expression of connective tissue growth factor (CTGF).** Correlations between mRNA levels of TG2 following individual knockdown and mRNA levels of profibrotic, proliferation and apoptotic markers in neonatal rat ventricular fibroblasts (n = 18): (A) collagen 3A1 (COL 3A1), (B) collagen 1A1 (COL 1A1), (C) fibronectin 1 (FN 1), (D) CTGF, (E) matrix metalloproteinase-2 (MMP-2), (F) MMP-9, (G) α-smooth muscle actin (α-SMA), (H) transforming growth factor β1 (TGF-β1), (I) connexin 43 (CX 43), (J) periostin, (K) BCL-2-associate X protein (BAX), (L) B-cell lymphoma 2 (BCL-2), (M) BAX/BCL-2 ratio, (N) cyclin D1 (CCND 1), and (O) cyclin E2 (CCNE 2). $R^2$: Correlation coefficient; in the equations, (X) and (Y) respectively denote TG2 expression and expression of the profibrotic, proliferation or apoptotic markers.

completely unexpected as siRNA can reduce gene expression in genes other than those targeted [30]. siRNAs can tolerate multiple mismatches on the target mRNA, resulting in the formation of incomplete complementary pairs that use the silencing effect of their gene [30]. siRNAs have been shown to influence the translation of unintentional transcripts with partial complementarity [31, 32]. As a result, each siRNA for a specific target has the same on-target action but different off-target activity. In the present work, no expression of TG3, TG4, and TG6 was detected before and after transfection with TG1 or TG2, suggesting that TG1 or TG2 knockdown did not lead to compensatory activation of the other isoforms. In contrast, a few reports indicated that total TG activity remained nearly constant in the cardiac tissues of TG2 knockout mice and was concomitant to increased transcriptional expression of TG3, TG5, and TG6, raising the possibility that other TG isoforms could compensate for decreased activity of TG2 [33, 34]. The possible overlap and interplay between the expression and functions of these TGs in cardiac cells warrant specific investigation.

The choice of a neonatal rat model in the present study is highly relevant due to the well-established advantages of using juvenile rodents to investigate cardiac diseases and to the distinct advantages of rat models. Neonatal rat and murine models have both been used to investigate physiological responses in health and disease [35]. Advantages of rodents include their high reproductive capacity, short life cycle, ease of maintenance in controlled facilities, as well as abundant gene pools [36–38]. Rats are larger than mice, allowing for larger tissue samples to be collected as well as savings in labor and research costs [39, 40]. Rats being one of the most commonly used animals in biomedical research, their physiology is well known and similar enough to human physiology to generate relevant experimental data that can be applied clinically [37, 39, 40]. According to Iannaccone and Jacob [37] and Abbott [38], rats are an excellent model for cardiovascular diseases in humans enabling research with diverse genetic stocks. While gene targeting technologies for gene deletion and manipulation have proved challenging to implement in rats, the latest innovations in transgenic technology and significant advances in mouse genetic toolboxes have laid the ground for similar developments to become applicable in rats [37, 38, 41, 42].

Despite the well-recognized pleiotropic nature of TGs, most studies investigating the role of TGs in cardiac physiopathology have focused solely on TG2. While a few reports pointed to a high expression of TG2 in cardiac fibroblasts and suggested that this was the predominant isoform in fibroblasts [19, 23, 43], which is in agreement with our findings, our study further showed that TG1 was also expressed, both in cardiomyocytes and in fibroblasts. Yet, to date, there are hardly any studies on the functions of TG1 in these cells. In contrast, mounting evidence from *in vivo* studies suggests functional and cellular signalling roles for TG2 in cardiac tissue remodelling processes and their dysregulation in myocardial hypertrophy and infarction and heart failure, among other cardiac diseases [17, 19, 44–46]. Increased expression of this isoform has been linked to excessive deposition of insoluble collagen, a key factor in the development of cardiac fibrosis [20]. According to Wang et al. [17], TG2 stimulates cardiac fibroblasts and excessive ECM deposition, resulting in cardiac fibrosis. Our findings support this view as the silencing of TG2 or TG1 decreased the amount of insoluble collagen and the collagen cross-linking ratio by more than half in fibroblasts. Collagen cross-linking also decreased upon TG2 knockout in the mouse model of pressure-overloaded heart [47]. Similarly, pharmacological inhibition of TG2 activity was found to reduce hypoxia-induced cross-linking activity in pulmonary hypertension [23].

We found that COL 1A1 and COL 3A1 transcription in fibroblasts remained unchanged after TG2 knockdown, while TG1 silencing depressed COL 3A1 transcript levels. Moreover, TG1 and COL 1A1 expression were positively correlated. In partial agreement with our findings, Shinde et al. [48] found that collagen transcription was not significantly affected by the

addition of exogenous recombinant TG2 to fibroblasts *in vivo*. In contrast, Penumatsa et al. [23] showed that reduced TG2 activity induced by TG inhibitor ERW1041E inhibited hypoxia-induced transcription of collagen 1 and α-SMA in right ventricular fibroblasts. In our work, α-SMA expression was enhanced upon TG2 silencing, suggesting the expression of this ECM protein may be repressed by TG2 under nonpathological conditions. Increased expression of α-SMA is recognized as a characteristic of fibroblasts differentiation into highly specialized myofibroblasts involved in tissue repair following cardiac injury [49]. Under normal conditions, TG2 repressing action on α-SMA may therefore protect against unnecessary tissue repair and excessive EMC deposition. No change in MMP-2 and MMP-9 expression was detected upon knocking down TG1 or TG2. However, positive associations were found between MMP-2 and TG1 expression and between those of MMP-9 and TG2. In previous reports, increased TG2 expression was associated with increased MMPs and collagen during aging [46], while in the pressure-overloaded myocardium, lower levels of MMP-2 and MMP-9 expression have been reported in TG2 knockout hearts as compared to wild-type cells [19].

TGF-β1 transcription was enhanced upon silencing TG2 or TG1 and was positively correlated with TG2 expression but inversely correlated with TG1. These results are in partial agreement with Johnson et al. [50]'s findings of a higher conversion rate of latent TGF-β upon TG2 inhibition in renal tissues. However, in the pressure-overloaded myocardium, TG2 did not modulate TGF-β1-mediated responses in cardiac fibroblasts [19]. While the current view indicates that TG2 is implicated in controlling TGF-β1 signalling and activation of latent TGF-β1 in cardiac tissues [51], the interplay between this growth factor and the expression of TG2 and TG1 remains elusive. Addition of TGF-β1 to cardiac fibroblasts was found to induce the synthesis of TG2 but not that of TG1 and other TGs [19]. Transcription of CTGF was depressed in our study upon silencing TG1 or TG2, suggesting that these isoforms may promote CTGF synthesis in cardiac fibroblasts under normal conditions. Depressed CTGF transcription is consistent with the reduced transcription of COL 3A1 upon TG1 silencing as CTGF stimulates the synthesis of ECM proteins, mainly collagen, which contributes to cardiac fibrosis [52, 53]. The positive association between CTGF and TG2 expression in our work is consistent with the positive correlation between TG2 and CTGF protein levels in the renal fibrotic model [54].

Aside from regulating cell growth and differentiation, TG2 has also been studied for its likely implication in regulating cell proliferation and survival in different tissues [55–57]. In the present work, we showed that TG2 knockdown increased the proliferation of cardiac fibroblasts and the transcription of CCND 1 and that TG2 expression was positively correlated with CCND 1, while TG1 expression was inversely associated with CCNE 2 transcription. Cyclin-dependent kinases such as CCND 1 and CCNE 2 are crucial for regulating the proliferation and cell cycle [58]. Consistent with our findings, TG2 was found to induce a matrix-preserving role and to restrain the proliferation of cardiac fibroblasts by promoting the synthesis of tissue inhibitors of metalloproteinases (TIMP) [48]. Moreover, cardiac fibroblasts from TG2 knockout mice displayed an increased proliferative capacity [19]. In breast cells, by contrast, Xu et al. [59] concluded that TG2 stimulates cancerous tumor cell proliferation and increases glycolysis. Our findings of decreased CTGF expression and increased fibroblast proliferation induced by TG2 silencing are consistent with the results of Zhen et al. [60] who showed that decreased expression of CTGF promoted cell cycle progression and proliferation through FAK/PI3K/AKT, epithelial-mesenchymal transition and MMP pathways in nasopharyngeal carcinoma. They further showed that CTGF knockdown activated the oncogenic cell cycle regulator pRB (Ser 780), which includes CCND 1 [60]. CTGF expression seems to be associated with the development of different types of cancer [61, 62] and its reduction has proved beneficial for tumor progression and prognosis of certain cancers [63–65]. While the role of CTGF in fibrosis remains controversial, there is some evidence that CTGF expression is essential for

fibroblast proliferation and that CTGF inhibition may help prevent and reverse the fibrosis process [16, 66].

No significant change in fibroblast expression of apoptotic markers was found in the present study. However, TG1 transcription levels were significantly associated with BCL-2 expression and the BAX/BCL-2 ratio, while TG2 displayed a positive association with BAX and BCL-2 expression, pointing to a possible role of both isoforms in promoting apoptosis in fibroblasts. To date, TG2 is the isoform that was most frequently linked to apoptosis in cardiac tissues. Its colocalization with apoptotic markers, which has been reported in cardiomyocytes [67], is suggestive of a signalling action in apoptotic pathways [68, 69]. In other fibrotic conditions, apoptotic cells were found to release TG2 [70], while the rate of apoptosis in human promonocytic cells was reduced by silencing TG2 [71]. According to available evidence, cardiac remodelling involves fibroblast proliferation and differentiation, as well as cardiomyocyte hypertrophy and apoptosis [72, 73]. In cardiomyopathic patients, NIX, a BCL-2 family member that stimulates cardiomyocyte apoptosis, has been identified as a contributor to cardiac fibrosis development [74]. Overexpression of TG2 in the heart was associated with cardiomyocyte apoptosis, structural changes, detrimental hemodynamic modifications, as well as cardiac fibrosis and hypertrophy [75, 76]. In the neonatal rat model, TG2 was shown to promote cardiomyocyte apoptosis under oxidative stress by inducing $Ca^{2+}$ overload via PKC- and PLC-$\delta$1-dependent signalling pathways, while its silencing protected the cells from oxidative stress-induced apoptosis [24].

While the inhibition of TG2 and other isoforms may offer protection against cardiac fibrosis under certain conditions, their protective functions should also be considered when designing anti-fibrotic therapies involving TGs [21, 51]. As pointed out by Griffin et al. [22], Al-U'datt et al. [21], and Szondy et al. [51], the inhibition of TG2 to reduce the fibrotic response may compromise tissue repair and worsen certain pathologies, as in cardiac fibrosis associated with factor XIII-A deficiency. Likewise, TG2 appears to protect cardiomyocytes against heart ischemia/reperfusion injury and to attenuate ischemic-induced cell death by modulating specific transcriptional processes [77, 78]. Protection against Fas-mediated cell death and CCl4-induced liver injury was also evidenced in hepatocytes [79, 80], while a growing body of evidence indicates that TG2 can both promote and inhibit apoptosis [81–83]. An additional, presumably protective function of TG2, proposed by Fésüs and Szondy [81], involves the prevention of inflammation, tissue injury and autoimmunity once apoptosis has been initiated. According to these authors, this function of TG2 is partially achieved by being expressed and activated in macrophages that digest apoptotic cells and by mediating cross-talks between dying cells and phagocytes (neutrophiles and monocytes) [81]. During myocardial infraction, cardiomyocyte necrosis induces a strong inflammatory response that is important for cardiac repair [84]. In the inflammatory phase, signals released by necrotic cardiomyocytes induce the production of cytokines and chemokines that mediate the recruitment monocytes, lymphocytes, and neutrophils in the infarcted myocardium [84]. TG2 and factor XIII-A are both found in monocytes and macrophages [85] and mediate the adhesion and extravasation of monocytes. TG2 is also required for monocyte maturation and activation into macrophages [86]. More research is needed to better understand the opposing roles of TG2 and other isoforms, either inhibiting or promoting the fibrotic response in different cardiac pathologies.

## Study limitations

The *in vitro* model used in this study enabled a comprehensive investigation of TG1 and TG2 contribution to fibrotic signalling, collagen cross-linking and cell proliferation in rat ventricular fibroblasts. The main limitations include the limited specificity of siRNA for TG1 silencing

and the lack of validation of mRNA expression at the protein level. As TG1 silencing also depressed TG2 transcription, the results obtained for TG1 should be interpreted with caution. By contrast, TG2 knockdown was specific to this isoform in fibroblasts. The linear correlations evidenced in our work should also be interpreted cautiously as they carry several imitations inherent to this type of statistical analysis. For example, the existence of subgroups within our dataset may have led to a false sense of relationship overall, even if no association existed within each subgroup [87]. An additional limitation is related to the relatively small number of observations available for the correlations analyses reported. These associations should be studied further with methodologies permitting more definitive conclusions regarding TG1 and TG2 roles and modes of action.

Further research using genetic inhibition of TG1 and TG2 in cardiac cells is also necessary to corroborate the present findings and ascertain the mechanistic basis of TGs implication in cardiac fibrotic processes. Moreover, the use of specific TG inhibitors should prove valuable to validate the results through *in vivo* studies. Analyses at the protein level combined with functional assays, such as cell contractility, calcium levels, and direct measures of cell proliferation and apoptosis (e.g., thymidine incorporation assay and apoptotic index by immunohistochemistry), are warranted to confirm the biological functions and mechanisms of action of TG1 or TG2 in cardiac fibroblasts and myocytes. LOX isoforms could also be assessed for their roles and potential clinical applications in the early diagnosis and management of heart diseases. Finally, the fact that this study was conducted *in vitro* does not allow any prediction over the roles that TG1 and TG2 may play in the host cardiac cells. The therapeutic potential of siRNA in humans is limited at present due to pharmacokinetic challenges, delivery and bioavailability issues as well as off-target effects [88]. These hurdles necessitate specific chemical alterations to either siRNA or the delivery system before siRNA-based therapeutics can be widely deployed in clinical settings [88]. We anticipate that the present findings will pave the way for future translational research on the therapeutic benefits and safety of TG inhibition in cardiac diseases.

## Conclusion

This study showed that TG1 and TG2 were both expressed in neonatal rat ventricular cardiomyocytes and fibroblasts. Silencing TG1 or TG2 elicited significant changes in the transcript levels of fibrotic markers COL 3A1, α-SMA, CTGF, TGF-β1, and CCND 1 in fibroblasts, while increasing fibroblast proliferation and reducing the amount of insoluble collagen and collagen cross-linking in these cells. Strong associations between TG1 or TG2 transcript levels and those of several profibrotic, proliferation, and apoptotic markers, including COL 1A1, MMP-2, CTGF, CCNE 2, and BAX/BCL-2, were also evidenced, lending further support to a possible role of TG1 and TG2 in mediating certain fibrotic signalling processes. No compensatory expression of TG3, TG4 or TG6 was detected upon TG1 or TG2 knockdown. These results suggest that TG1 and TG2 are both implicated in collagen cross-linking and in cellular signalling processes underlying ECM synthesis and degradation, cell proliferation and apoptosis. Additional pathway exploration studies are warranted to establish the mechanistic pathways of TG1 and TG2 actions in cardiac cells during normal and pathological remodelling of the ECM, which will help inform the selection of specific TG targets for therapeutic intervention. While several TG inhibitors have emerged as candidate drugs for cardiac diseases, they have not yet been tested in clinical trials specific to cardiac diseases [21, 51]. The action of TGs in cardiac cells must be more fully elucidated to enable further innovations for safe, reliable and effective pharmacological interventions in this area.

## Supporting information

**S1 Table. Oligonucleotide sequences of custom-designed primers for quantitative PCR.**
(DOCX)

**S1 Graphical abstract.**
(PPTX)

## Author Contributions

**Conceptualization:** Doa'a G. F. Al-U'datt, Carole C. Tranchant, Jenan Alqbelat.

**Data curation:** Doa'a G. F. Al-U'datt, Belal Al-Husein, Roddy Hiram, Ahmed Al-Dwairi, Mohammad AlQudah, Othman Al-shboul, Jenan Alqbelat.

**Formal analysis:** Doa'a G. F. Al-U'datt, Belal Al-Husein, Roddy Hiram, Jenan Alqbelat.

**Funding acquisition:** Doa'a G. F. Al-U'datt, Belal Al-Husein.

**Investigation:** Doa'a G. F. Al-U'datt, Belal Al-Husein, Ahmed Al-Dwairi, Mohammad AlQudah, Othman Al-shboul, Jenan Alqbelat.

**Methodology:** Doa'a G. F. Al-U'datt, Carole C. Tranchant, Belal Al-Husein, Ahmed Al-Dwairi, Mohammad AlQudah, Othman Al-shboul, Jenan Alqbelat.

**Project administration:** Doa'a G. F. Al-U'datt.

**Resources:** Doa'a G. F. Al-U'datt, Carole C. Tranchant, Belal Al-Husein, Ahmed Al-Dwairi, Mohammad AlQudah, Othman Al-shboul, Saied Jaradat, Ali Almajwal.

**Software:** Doa'a G. F. Al-U'datt, Roddy Hiram, Saied Jaradat.

**Supervision:** Doa'a G. F. Al-U'datt, Belal Al-Husein.

**Validation:** Doa'a G. F. Al-U'datt, Carole C. Tranchant, Jenan Alqbelat.

**Visualization:** Doa'a G. F. Al-U'datt, Carole C. Tranchant, Roddy Hiram, Jenan Alqbelat.

**Writing – original draft:** Doa'a G. F. Al-U'datt, Jenan Alqbelat.

**Writing – review & editing:** Doa'a G. F. Al-U'datt, Carole C. Tranchant, Belal Al-Husein, Roddy Hiram, Ahmed Al-Dwairi, Mohammad AlQudah, Othman Al-shboul, Saied Jaradat, Ali Almajwal.

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
