## [Decision Letter · Decision Letter 0]

1 Dec 2022

PONE-D-22-29124Targeting the transglutaminase 1 (TG1) and TG2 signalling cascades in cardiac fibrosis: extra- and intra-cellular roles in fibroblastsPLOS ONE

Dear Dr. Al-u'datt,

Thank you for submitting your manuscript to PLOS ONE. After careful consideration, we feel that it has merit but does not fully meet PLOS ONE’s publication criteria as it currently stands. Therefore, we invite you to submit a revised version of the manuscript that addresses the points raised during the review process.

We look forward to receiving your revised manuscript.

Kind regards,

Michael Bader

Academic Editor

PLOS ONE

Journal Requirements:

"The authors extend their appreciation to the Researchers Supporting Project number (RSP2022R502), King Saud University, Riyadh, Saudi Arabia for funding this project."

Reviewers' comments:

Reviewer's Responses to Questions

**Comments to the Author**

1. Is the manuscript technically sound, and do the data support the conclusions?

Reviewer #1: No

Reviewer #2: No

2. Has the statistical analysis been performed appropriately and rigorously? 

Reviewer #1: No

Reviewer #2: Yes

3. Have the authors made all data underlying the findings in their manuscript fully available?

Reviewer #1: Yes

Reviewer #2: Yes

4. Is the manuscript presented in an intelligible fashion and written in standard English?

Reviewer #1: Yes

Reviewer #2: Yes

5. Review Comments to the Author

Reviewer #1: In this paper, Ali et al. comprehensively investigate the relationship between transglutaminase 1 and 2 at the transcriptome level and the proliferation, apoptosis and fibronectin-related marker of rat ventricular fibroblasts. The authors inferred that TG1 and TG2 play a role in cardiac fibrosis by interfering with the biological functions of fibroblasts through a variety of enzymes and growth factors through siRNA silencing and expression level correlation analysis. tG1 and TG2 may be targets for the study and treatment of cardiac fibrosis. Overall the authors' study is comprehensive, but numerous innovative and rigorous flaws prevented the publication of this paper, with the following comments.

Major comments：

1.As it was mentioned in your paper that TG family has numeric isomers, from TG1 to TG7 and so on, the reason why only TG1, and TG2 were selected for research requires an explanation.

2. The scientific meaning of the conclusion in this study should be discussed. (for targeted therapy or pathway exploration? Is there any clinical trial carrying on? If not, where the limitation existed?)

3. the study seriously lacks the validation of protein level, as it is widely recognized that Analysis at the transcriptional level alone may not be able to characterise the biological function of the cell. By the way, in vivo study can also contribute to the enhance the persuasiveness of the results.

4. the choice of rat instead of mice should be discussed, and the advantages or limitations worth to be mentioned in the comparison between two models.

5. The role of apoptosis in the cardiac fibrosis should be discussed separately. In fibroblasts proliferation and secretion of associated proteins should be the main factors promoting fibrosis, whereas apoptosis may play more of a fibrosis-promoting role in cardiomyocytes. In the article the authors examine apoptosis in fibroblasts and thus have to make a specific discussion as to what role proliferation and apoptosis play respectively in fibroblasts and which plays a more dominant role?

6. Some of the factors of interest are significant while some are not. Those that are significant were stated as "consistent with previous" but those that are not should also be interpreted accordingly.

7. most of this work are repetitive and validation experiments of previous study, the novelty of this study worth to be mentioned specifically.

8. CTGF is a pro-proliferative factor, but its expression is reduced after silencing TGs, this controversy should be explained.

9. Limitations of this study need to be written more clearly.

Minor comments:

1. the “while” in the sentence in 3.1 is incorrectly capitalised.

2. the font in the table is not uniform.

3. the author's contribution could be mentioned.

4. the correlation fit is not good enough, the number of points is small and in some of the plots there are obvious subgroups, the fit may be a bit too far-fetched and this needs to be discussed and explained.

5. In the previous study, TG2 was also shown to have a protective function, which should also be taken into account.

6. the apoptotic index and marker could be more comprehensive.

7. graphical abstract should be mentioned as the overall description of the pathway is cluttered.

8. what role in the monocyte/macrophage and in cardiomyocytes could be further explored and discussed.

Reviewer #2: In this study, the role of TG2 and TG2 signaling cascades in cardiac fibrosis was to be tested. However, in its present form, the manuscript is too preliminary to support any functional or mechanistic role of TG2 in cardiac fibroblasts.

• Were isolated fibroblasts authenticated? If so, how?

• For siRNA – please document the siRNA used specifically, and the reagent used to transfect the cells. More details are needed in the methods – were cells serum starved? If so, for how long? In what media were they incubated after transfection?

• In Fig 1A,B: In the fibroblasts, siTG2 yields better TG1 knockdown than the TG1 siRNA. It is unclear what levels of TG1/tTG expression there is at baseline (neg siRNA). As the TG2 siRNA knocks down both TG1 and tTG, how does this allow any distinction between the two?

• Was compensatory expression of other TGases evaluated upon knockdown? In the TG2-/- mouse, there is compensation from other TGases in the cardiovascular system.

• The axis labels and titles are illegible in Figs 3, 4, 5, 6.

• The purpose of the correlation analysis in Figs 5, 6 is unclear. – how does this analysis support or refute any core function of TG2?

• The purpose of hydroxyproline content is also unclear – these are generated by LOX enzymes. For TGases it would make more sense to look for TG-crosslinks (isopeptide bond), for which there is an antibody available.

• Some additional mechanistic or functional assays illustrating changes in cardiac fibroblast signaling or behavior would be useful (eg wound healing, motility, proliferation).

• A more stringent set of experiments are needed to support the notion that TG1 and TG2 stimulate/regulate enzymes and growth factors. A simple correlation analysis does not support this.

Minor:

• Overall the manuscript is well written, however some minor English language editing is needed. Eg. In methods: “heart pups” line 85.

6. PLOS authors have the option to publish the peer review history of their article (what does this mean?). If published, this will include your full peer review and any attached files.

Reviewer #1: No

Reviewer #2: No

---

## [Author Response · Author response to Decision Letter 0]

11 Jan 2023

Manuscript PONE-D-22-29124 R1

Revised title: “Involvement and possible role of transglutaminases 1 and 2 in mediating fibrotic signalling, collagen cross-linking and cell proliferation in neonatal rat ventricular fibroblasts”

Initial title: “Targeting the transglutaminase 1 (TG1) and TG2 signalling cascades involved in cardiac fibrosis: Extra- and intra-cellular roles in fibroblasts”

Authors Response to Editor’s and Reviewers’ Comments – December 2022

Response to Editor’s comments

Dear Editor,

Thank you kindly for your thorough assessment of our work. Your constructive comments are appreciated. They have been considered to improve the manuscript as indicated below. All the comments and suggestions from the Reviewers have also been addressed. Track changes (yellow font) are used to highlight the revisions in the submitted version (R1).

Please let us know if you have any queries.

Best regards

Comments/Authors’ response

Comment 1. Please ensure that your manuscript meets PLOS ONE's style requirements, including those for file naming. The PLOS ONE style templates can be found at 

Response We have revised our manuscript thoroughly to meet PLOS ONE’s style requirements, including those for file naming, as outlined in the above guidelines.

Comment 2. To comply with PLOS ONE submissions requirements, in your Methods section, please provide additional information regarding the experiments involving animals and ensure you have included details on (1) methods of sacrifice, (2) methods of anesthesia and/or analgesia, and (3) efforts to alleviate suffering.

Response Additional details regarding the experiments involving animals have been added to the Methods, as suggested, including methods of anesthesia, methods of sacrifice, as well as efforts to alleviate suffering. Briefly, neonatal rat pups were separated from their mother and were pre-anesthetized by hypothermia (indirect contact with ice as the pups were wrapped in gaze) to lessen any stress on them. Their abdomen was sterilized with 70% ethanol for local anesthesia before they were euthanized by decapitation with sharp surgical scissors by a trained researcher. Their chest was cut open with sharp scissors and the beating heart was quickly excised then placed in HBSS (sterile calcium- and magnesium-free Hank’s Balanced Salt Solution) medium for isolation of fibroblasts and myocytes. All the animal handling procedures were approved by the Animals Research Ethics Committee of Jordan University of Science and Technology.

See manucript, Lines 117-124.

Comment 3. We note that the grant information you provided in the ‘Funding Information’ and ‘Financial Disclosure’ sections do not match. When you resubmit, please ensure that you provide the correct grant numbers for the awards you received for your study in the ‘Funding Information’ section.

4. Thank you for stating the following financial disclosure: "The authors extend their appreciation to the Researchers Supporting Project number (RSP2022R502), King Saud University, Riyadh, Saudi Arabia for funding this project." Please state what role the funders took in the study. If the funders had no role, please state: "The funders had no role in study design, data collection and analysis, decision to publish, or preparation of the manuscript." If this statement is not correct you must amend it as needed. Please include this amended Role of Funder statement in your cover letter.

Response The funding information and financial disclosure sections were revised as follows:

Funding: This study was supported by Jordan University of Science and Technology Deanship of Research Researchers Supporting Project number (RSP2022R502); and King Saud University, Riyadh, Saudi Arabia. The funders had no role in the study design, data collection and analysis, decision to publish, or preparation of the manuscript.

Competing interests: The authors declare that they have no competing interests.

See manucript, Lines 483-488.

Response to Reviewers’ comments

Comments/Authors’ response

Comment 1. Is the manuscript technically sound, and do the data support the conclusions? The manuscript must describe a technically sound piece of scientific research with data that supports the conclusions. Experiments must have been conducted rigorously, with appropriate controls, replication, and sample sizes. The conclusions must be drawn appropriately based on the data presented.

Reviewer #1: No

Reviewer #2: No

Response We have addressed all the Reviewers’ comments to improve our manuscript making sure to provide sufficient experimental details and to present conclusions that are based on the data obtained. The changes made to each section are presented in the individual responses to each Reviewer and to the Editor. The main changes and improvements include:

- The novelty of the study has been highlighted in the Abstract, Introduction and Discussion.

- The literature gaps and conflicting evidence addressed in this paper were made more explicit in these sections.

- Study limitations are clearly acknowledged at the end of the discussion.

- Additional details about the control (negative silencer control (NegsiRNA)), replication and sample sizes have been provided in the Methods and figure legends. 

- The conclusion is based on the results presented. It better articulates what the study is adding to existing evidence and offers suggestions for additional research.

Comment 2. Has the statistical analysis been performed appropriately and rigorously?

Reviewer #1: No

Reviewer #2: Yes

Response We have carefully revised our manuscript to address Reviewers’ comments regarding correlation analysis (Fig. 5 and 6). The changes made are summarized in the individual responses to each Reviewer. Moreover, additional details about the control, replication and sample sizes have been provided in the Methods and figure legends. 

Comment 3. Have the authors made all data underlying the findings in their manuscript fully available? The PLOS Data policy requires authors to make all data underlying the findings described in their manuscript fully available without restriction, with rare exception (please refer to the Data Availability Statement in the manuscript PDF file). The data should be provided as part of the manuscript or its supporting information, or deposited to a public repository. For example, in addition to summary statistics, the data points behind means, medians and variance measures should be available. If there are restrictions on publicly sharing data those must be specified. 

Reviewer #1: Yes

Reviewer #2: Yes

Response Thank you.

Response to Reviewer 1’s comments

Dear Reviewer,

Thank you kindly for your thorough assessment of our work. Your constructive comments are appreciated. They have been considered to improve the manuscript as indicated below. All the amendments are highlighted (in yellow) in the revised manuscript. 

Please let us know if you have any queries.

Best regards 

Comments/Authors’ response

Comment Reviewer #1: In this paper, Ali et al. comprehensively investigate the relationship between transglutaminase 1 and 2 at the transcriptome level and the proliferation, apoptosis and fibronectin-related marker of rat ventricular fibroblasts. The authors inferred that TG1 and TG2 play a role in cardiac fibrosis by interfering with the biological functions of fibroblasts through a variety of enzymes and growth factors through siRNA silencing and expression level correlation analysis. tG1 and TG2 may be targets for the study and treatment of cardiac fibrosis. Overall the authors' study is comprehensive, but numerous innovative and rigorous flaws prevented the publication of this paper, with the following comments.

Major comments:

1. As it was mentioned in your paper that TG family has numeric isomers, from TG1 to TG7 and so on, the reason why only TG1, and TG2 were selected requires an explanation.

Response We have addressed all of your comments to improve our manuscript. Special consideration was given to better articulate the novelty of the study and improve the rigour of our report, as suggested. The changes made are summarized in this table.

1. The expression of several isoforms, namely TG1, TG2, TG3, TG4, and TG6, was assessed by qPCR in the present study, as indicated in the revised Methods and Results. This is part of the originality of the study. We found that TG1 and TG2 were the only isoforms expressed in neonatal rat ventricular fibroblasts and cardiomyocytes, with TG2 being the predominant isoform before transfection. The other isoforms were not detected before and after transfection. Cycle threshold (Ct) values were therefore only calculated for TG1 and TG2.

TG2 was selected due to its suspected, yet little understood functions in cardiac diseases and fibrosis (Al-U’datt et al., 2022), as indicated in the Introduction. Other TG isoforms, including TG1, may have functions relevant to cardiac pathophysiology (Sane, Kontos, & Greenberg, 2007). TG1 expression was found in endothelial cells (EC) (Baumgartner et al., 2004) but has received far less attention than TG2 in relation to cardiac fibrosis. As indicated in the Introduction, other TGs were measured in our work to determine whether TG1 or TG2 knockdown may lead to compensatory activation of other TG isoforms.

Comment 2. The scientific meaning of the conclusion in this study should be discussed (for targeted therapy or pathway exploration? Is there any clinical trial carrying on? If not, where the limitation existed?)

Response The scientific meaning of the conclusion and the clinical implications of our findings have been better articulated, as suggested. As noted in the conclusion, these findings will provide the basis for future gene knockdown experimentation and pathway exploration studies. They will also inform the selection of specific TG targets for TG inhibition at target sites and therapeutic gene knockdown (siRNA-based therapeutics). While several TG inhibitors have emerged as candidate drugs for cardiac diseases, they have not yet been tested in clinical trials specific to cardiac diseases. The action of TGs in cardiac diseases and fibrosis must be more fully elucidated to enable further innovations for safe, reliable and effective pharmacological interventions in this area.

See manucript, Lines 431-478.

Comment 3. the study seriously lacks the validation of protein level, as it is widely recognized that Analysis at the transcriptional level alone may not be able to characterise the biological function of the cell. By the way, in vivo study can also contribute to the enhance the persuasiveness of the results.

Response We agree and have discussed this as a limitation in our revised manuscript. An in vivo study with assessment of TG2 levels was recently conducted by our lab to investigate the role of TG2 in the heart failure rat model. Findings from this distinct study are presented in a separate manuscript due to the large amount of data; they are under review in another journal.

Comment 4. the choice of rat instead of mice should be discussed, and the advantages or limitations worth to be mentioned in the comparison between two models.

Response For the type of experiments involved in our research and presented in this report, rats have several advantages over mice and larger animal models. A discussion of the considerations underlying the choice of animal model (rats vs. mice) and of their respective advantages and limitations has been added to the Discussion section as suggested.

See manuscript, Lines 311-324.

Comment 5. The role of apoptosis in the cardiac fibrosis should be discussed separately. In fibroblasts proliferation and secretion of associated proteins should be the main factors promoting fibrosis, whereas apoptosis may play more of a fibrosis-promoting role in cardiomyocytes. In the article the authors examine apoptosis in fibroblasts and thus have to make a specific discussion as to what role proliferation and apoptosis play respectively in fibroblasts and which plays a more dominant role?

Response We appreciate the reviewer’s suggestion. We have added a more pointed discussion of the role that proliferation and apoptosis may play in fibroblasts and cardiomyocytes, respectively, and of the possible predominance of certain factors in these cells.

See manuscript, Lines 398-430.

Comment 6. Some of the factors of interest are significant while some are not. Those that are significant were stated as "consistent with previous" but those that are not should also be interpreted accordingly.

Response We added an interpretation of the findings for each factor in light of previously published data (whenever available). We highlighted both agreement and conflicting information between our findings and earlier reports when available.

See manuscript, Lines 345-347, 359-361 and 380-391.

Comment 7. most of this work are repetitive and validation experiments of previous study, the novelty of this study worth to be mentioned specifically.

Response We highlighted the novelty of our study in the Introduction and Discussion sections, while avoiding repetitive sentences. The literature gaps and conflicting evidence addressed in this work were also made more explicit in these sections.

To our knowledge, the present study is the first to investigate the expression of several TG isoforms (TG1, TG2, TG3, TG4 and TG6) in cardiac fibroblasts and their possible actions in cardiac fibrosis-promoting signalling cascades through siRNA-mediated knockdown. While the involvement of TG2 in cardiac fibrosis is well documented, its roles remain poorly understood. TG isoforms other than TG2 may also have functions relevant to cardiac pathophysiology. However, these isoforms have received far less attention than TG2 in relation to cardiac fibrosis; their expression in cardiac cells has seldom been documented.

See manuscript, Lines 96-110.

Comment 8. CTGF is a pro-proliferative factor, but its expression is reduced after silencing TGs, this controversy should be explained.

Response The controversy over the proliferative role of CTGF has been addressed in the Discussion. It may reflect the complex biology of this matricellular protein and its ability to modulate diverse signalling pathways leading to cell adhesion and migration, myofibroblast activation, as well as ECM deposition and remodeling (Lipson, Wong, Teng, & Spong, 2012). While the role of CTGF in fibrosis remains controversial, evidence suggests that CTGF expression is essential for fibroblast proliferation and that CTGF inhibition may help prevent and reverse the process of fibrosis (Lipson et al., 2012). CTGF is also commonly used as a profibrotic marker in fibrotic diseases (Effendi & Nagano, 2022; Morales et al., 2018). Selective expression of CTGF in fibroblasts was found to promote systemic tissue fibrosis in vivo (Sonnylal et al., 2010). Our findings of decreased CTGF mRNA after silencing TG1 or TG2 suggest a modulatory role of both TGs on CTGF expression.

See manuscript, Lines 379-392.

Comment 9. Limitations of this study need to be written more clearly.

Response Study limitations are now clearly acknowledged in a distinct paragraph at the end on the discussion. 

See manuscript, Lines 431-459.

Comment Minor comments:

1. the “while” in the sentence in 3.1 is incorrectly capitalised.

Response Thank you for bringing this to our attention. Typos have been corrected.

Comment 2. the font in the table is not uniform.

Response Table 1 and Table S1 have been thoroughly revised according to PLOS ONE’s guidelines. Uniform font (Times New Roman 12) is now used in the tables.

Comment 3. The author's contribution could be mentioned.

Response Authors’ contributions have been added at the end of the revised manuscript.

Comment 4. the correlation fit is not good enough, the number of points is small and in some of the plots there are obvious subgroups, the fit may be a bit too far-fetched and this needs to be discussed and explained.

Response We appreciate your suggestion. The limitations associated with correlation analysis are discussed at the end of the discussion.

See manuscript, Lines 437 to 443.

Comment 5. In the previous study, TG2 was also shown to have a protective function, which should also be taken into account.

Response TG2 protective function has been taken into account in the revised Discussion, as suggested.

See manuscript, Lines 409-430.

Comment 6. the apoptotic index and marker could be more comprehensive.

Response We agree that the use of an apoptotic index, combined with additional apoptotic markers, would have provided further evidence. A total of 15 markers were analysed in the present study, which focused on signalling pathways at the cellular level. It was not possible to include more markers due to financial constraints. This limitation has been acknowledged at the end of the discussion. Likewise, we have stressed the need for further research to better understand the roles of TGs, if any, in cardiac cell apoptosis. Immunostaining analyses could be recommended for apoptosis quantification at the tissue level.

One of the key unknowns is whether cardiac cell apoptosis plays a significant role in the pathogenesis of cardiac diseases. Due to the paucity of research addressing the possible roles of TGs other than TG2, we anticipate that the present findings will pave the way for future in vivo research to elucidate these roles in cardiac cells, which will assist in understanding TGs-mediated mechanistic pathways to cardiac fibrosis.

See manuscript, Lines 444-458.

Comment 7. graphical abstract should be mentioned as the overall description of the pathway is cluttered.

Response A graphical abstract was submitted with the revised manuscript as suggested.

Comment 8. what role in the monocyte/macrophage and in cardiomyocytes could be further explored and discussed.

Response Our study focused on fibroblasts and cardiomyocytes. Monocytes and macrophages were not studied. A discussion of the possible effect of TG2 on these cells has been added as suggested.

See manuscript, Lines 411-429.

Response to Reviewer 2’s comments

Dear Reviewer,

Thank you kindly for your thorough assessment of our work. Your constructive comments are appreciated. They have been considered to improve the manuscript as indicated below. All the amendments are highlighted (in yellow) in the revised manuscript. 

Please let us know if you have any queries.

Best regards

Comment Authors’ response

Comment Reviewer #2: In this study, the role of TG2 and TG2 signaling cascades in cardiac fibrosis was to be tested. However, in its present form, the manuscript is too preliminary to support any functional or mechanistic role of TG2 in cardiac fibroblasts.

• Were isolated fibroblasts authenticated? If so, how?

Response We have addressed all of your comments to improve our manuscript. The changes made are summarized in this table. 

• As indicated in the revised Methods, fibroblasts were authenticated using a validated method published in Heart Rhythm and Nature Communications (Duong, Xiao, Qi, & Nattel, 2017; Golden, Gollapudi, Gerilechaogetu, Li, Cristales, Peng, et al., 2012; Liu, Song, Qi, van Vliet, Xiao, Xiong, et al., 2021). This method, which was developed and validated in the lab of Dr. Nattel when first author conducted their PhD studies in Canada, enables the authentication of cell type by immunostaining fibroblast markers.

Comment • For siRNA – please document the siRNA used specifically, and the reagent used to transfect the cells. More details are needed in the methods – were cells serum starved? If so, for how long? In what media were they incubated after transfection?

Response We appreciate your suggestions. All the relevant details about cell culture and transfection have been added to the Methods. As indicated, the cells were starved for 24 h using a serum-deprived medium before transfection.

See manuscript, Lines 142 to 149.

Comment • In Fig 1A,B: In the fibroblasts, siTG2 yields better TG1 knockdown than the TG1 siRNA. It is unclear what levels of TG1/tTG expression there is at baseline (neg siRNA). As the TG2 siRNA knocks down both TG1 and tTG, how does this allow any distinction between the two?

Response mRNA expression levels in the three groups of cells were expressed relative to the values obtained in the control group (NegsiRNA) in order to show the change of gene expression relative to the control. As shown in Fig 1, siTG2 showed high specificity for TG2 in fibroblasts where it significantly reduced TG2 mRNA expression (Fig 1B) with no significant change in TG1 expression (Fig 1A). Clarifications have been added to the Methods and Discussion to address your comments. Study limitations are also discussed.

See manuscript, Lines 303-310, 431-459 and 469-478.

Comment • Was compensatory expression of other TGases evaluated upon knockdown? In the TG2-/- mouse, there is compensation from other TGases in the cardiovascular system.

Response The expression of five isoforms, namely TG1, TG2, TG3, TG4, and TG6, was assessed in the present study, as indicated in the revised Introduction and Results. We did not find any compensatory expression of TG3, TG4 and TG6 upon knockdown of TG1 or TG2. In contrast to TG1 and TG2, the other isoforms were not expressed at basal level. Their cycle threshold (Ct) values were therefore not calculated.

Comment • The axis labels and titles are illegible in Figs 3, 4, 5, 6.

Response We have increased the police size of the labels in Fig. 3, 4, 5 and 6, and have also revised the legends to clarify. The abbreviations used on the X- and Y-axes are now defined in the legends in addition to the text.

Comment • The purpose of the correlation analysis in Figs 5, 6 is unclear. – how does this analysis support or refute any core function of TG2?

Response Correlation analysis was used to assess the associations between TG1 or TG2 mRNA expression and the expression of the individual profibrotic, proliferation and apoptotic markers. While correlation does not indicate a direct effect, it can reveal meaningful relationships between some variables, which can be further investigated in future studies. This has been clarified in the Methods and Discussion. 

See manuscript, Lines 217-218, 341-347, 358-361, 370-371 and 373-377.

Comment • The purpose of hydroxyproline content is also unclear – these are generated by LOX enzymes. For TGases it would make more sense to look for TG-crosslinks (isopeptide bond), for which there is an antibody available.

Response Being a major component of collagen, hydroxyproline is routinely assessed to provide a measurement of collagen content (Hofman et al., 2011). In the present study, the concentration of hydroxyproline in the hydrolyzed fibroblast cells and supernatants was used to estimate the amount of insoluble collagen and soluble collagen, respectively. The ratio of insoluble to soluble collagen, an indication of the amount and possibly nature of cross-links present in collagen molecules (Wess et al., 1995), was also calculated. This clarification was added to the Methods.

LOX and TGs are both important contributors to enzymatic collagen cross-linking. Therefore, it is reasonable to expect that a change in TG activity could affect collagen composition. In fact, several studies have relied on the hydroxyproline assay to measure soluble and/or insoluble collagen in cells and tissues to assess the contribution of TG2 to changes in collagen composition (Popov, Sverdlov, Sharma, Bhaskar, Li, Freitag, et al., 2011; Shweke, Boulos, Jouanneau, Vandermeersch, Melino, Dussaule, et al., 2008; Steppan, Sikka, Jandu, Barodka, Halushka, Flavahan, et al., 2014). We agree that our findings could be corroborated in future studies through the use of an antibody technique.

See manuscript, Lines 186-189.

Comment • Some additional mechanistic or functional assays illustrating changes in cardiac fibroblast signaling or behavior would be useful (eg wound healing, motility, proliferation). • A more stringent set of experiments are needed to support the notion that TG1 and TG2 stimulate/regulate enzymes and growth factors. A simple correlation analysis does not support this.

Response To address your comments, the scope of our study has been better articulated in the title and research objectives. We also rephrased the conclusion, presented study limitations more clearly and added suggestions for future studies. 

Kindly note that the results of a proliferation assay were reported in Fig. 4. We agree that additional assays and experiments such as wound healing and motility assays are needed to elucidate the roles of TG1 and TG2 in cardiac cells. Thymidine incorporation, cell contractility, calcium levels, as well as immunostaining of apoptotic and proliferative markers could also be examined. As the present study was performed with a limited budget, the number of experiments had to be limited. Due to the paucity of research examining the roles of TG isoforms other than TG2, we anticipate that our findings will pave the way for future in vivo and in vitro research to elucidate these roles in cardiac cells, which will assist in understanding TGs-mediated mechanistic pathways to cardiac fibrosis.

See manuscript, Lines 444-459.

Comment Minor: • Overall the manuscript is well written, however some minor English language editing is needed. Eg. In methods: “heart pups” line 85.

Response Thank you for bringing this to our attention. The correction was made (“neonatal hearts”) and the manuscript was critically revised by native speaker and coauthor CCT who also contributed important intellectual content.

References cited in the response and/or revised manuscript 

Abbott, A. (2004). The Renaissance rat. Nature, 428(6982), 464-466.

Adam, O., Lavall, D., Theobald, K., Hohl, M., Grube, M., Ameling, S., Sussman, M. A., Rosenkranz, S., Kroemer, H. K., Schäfers, H.-J., Böhm, M., & Laufs, U. (2010). Rac1Induced Connective Tissue Growth Factor Regulates Connexin 43 and N-Cadherin Expression in Atrial Fibrillation. Journal of the American College of Cardiology, 55(5), 469-480.

Adam, O., Löhfelm, B., Thum, T., Gupta, S. K., Puhl, S. L., Schäfers, H. J., Böhm, M., & Laufs, U. (2012). Role of miR-21 in the pathogenesis of atrial fibrosis. Basic Res Cardiol, 107(5), 278.

Aggarwal, R., & Ranganathan, P. (2016). Common pitfalls in statistical analysis: The use of correlation techniques. Perspect Clin Res, 7(4), 187-190.

Al-U'datt D. G. F., Tranchant, C. C., Al-Dwairi, A., Alqudah, M., Al-Shboul, O., Hiram, R., Allen, B. G., Jaradat, S., Alqbelat, J., & Abu-Zaiton, A. S. (2022). Implications of enigmatic transglutaminase 2 (TG2) in cardiac diseases and therapeutic developments. Biochem Pharmacol, 201, 115104.

Alshaer, W., Zureigat, H., Al Karaki, A., Al-Kadash, A., Gharaibeh, L., Hatmal, M. M., Aljabali, A. A. A., & Awidi, A. (2021). siRNA: Mechanism of action, challenges, and therapeutic approaches. Eur J Pharmacol, 905, 174178.

Amat, S., Dahlen, C. R., Swanson, K. C., Ward, A. K., Reynolds, L. P., & Caton, J. S. (2022). Bovine Animal Model for Studying the Maternal Microbiome, in utero Microbial Colonization and Their Role in Offspring Development and Fetal Programming. Front Microbiol, 13, 854453.

Asif, M., Egan, J., Vasan, S., Jyothirmayi, G. N., Masurekar, M. R., Lopez, S., Williams, C., Torres, R. L., Wagle, D., Ulrich, P., Cerami, A., Brines, M., & Regan, T. J. (2000). An advanced glycation endproduct cross-link breaker can reverse age-related increases in myocardial stiffness. Proc Natl Acad Sci U S A, 97(6), 2809-2813.

Avery, N. C., & Bailey, A. J. (2005). Enzymic and non-enzymic cross-linking mechanisms in relation to turnover of collagen: relevance to aging and exercise. Scand J Med Sci Sports, 15(4), 231-240.

Braig, S., Wallner, S., Junglas, B., Fuchshofer, R., & Bosserhoff, A. K. (2011). CTGF is overexpressed in malignant melanoma and promotes cell invasion and migration. Br J Cancer, 105(2), 231-238.

Chang, C. C., Shih, J. Y., Jeng, Y. M., Su, J. L., Lin, B. Z., Chen, S. T., Chau, Y. P., Yang, P. C., & Kuo, M. L. (2004). Connective tissue growth factor and its role in lung adenocarcinoma invasion and metastasis. J Natl Cancer Inst, 96(5), 364-375.

Chen, H. (2018). Novel Rat Models for Atherosclerosis. Journal of Cardiology and Cardiovascular Sciences, 2, 29-33.

Cui, L., Zhang, Q., Mao, Z., Chen, J., Wang, X., Qu, J., Zhang, J., & Jin, D. (2011). CTGF is overexpressed in papillary thyroid carcinoma and promotes the growth of papillary thyroid cancer cells. Tumor Biology, 32(4), 721-728.

Diwan, A., Wansapura, J., Syed, F. M., Matkovich, S. J., Lorenz, J. N., & Dorn, G. W., 2nd. (2008). Nix-mediated apoptosis links myocardial fibrosis, cardiac remodeling, and hypertrophy decompensation. Circulation, 117(3), 396-404.

Duarte, L., Matte, C. R., Bizarro, C. V., & Ayub, M. A. Z. (2020). Transglutaminases: part I-origins, sources, and biotechnological characteristics. World J Microbiol Biotechnol, 36(1), 15.

Fésüs, L., & Szondy, Z. (2005). Transglutaminase 2 in the balance of cell death and survival. FEBS Letters, 579(15), 3297-3302.

Filiano, A. J., Tucholski, J., Dolan, P. J., Colak, G., & Johnson, G. V. W. (2010). Transglutaminase 2 protects against ischemic stroke. Neurobiology of Disease, 39(3), 334-343.

Frangogiannis, N. G., & Entman, M. L. (2005). Chemokines in Myocardial Ischemia. Trends in Cardiovascular Medicine, 15(5), 163-169.

Griffin, K. J., Newell, L. M., Simpson, K. R., Beckers, C. M. L., Drinkhill, M. J., Standeven, K. F., Cheah, L. T., Iismaa, S. E., Grant, P. J., Jackson, C. L., & Pease, R. J. (2020). Transglutaminase 2 limits the extravasation and the resultant myocardial fibrosis associated with factor XIII-A deficiency. Atherosclerosis, 294, 1-9.

Huang, G., Tong, C., Kumbhani, D. S., Ashton, C., Yan, H., & Ying, Q.-L. (2011). Beyond knockout rats: new insights into finer genome manipulation in rats. Cell cycle (Georgetown, Tex.), 10(7), 1059-1066.

Iannaccone, P. M., & Jacob, H. J. (2009). Rats! Disease Models & Mechanisms, 2(5-6), 206-210.

Johnson, T., Fisher, M., Haylor, J., Hau, Z., Skill, N., Jones, R., Saint, R., Coutts, I., Vickers, M., Nahas, A., & Griffin, M. (2008). Transglutaminase Inhibition Reduces Fibrosis and Preserves Function in Experimental Chronic Kidney Disease. Journal of the American Society of Nephrology : JASN, 18, 3078-3088.

Jyothirmayi, G. N., Soni, B. J., Masurekar, M., Lyons, M., & Regan, T. J. (1998). Effects of Metformin on Collagen Glycation and Diastolic Dysfunction in Diabetic Myocardium. J Cardiovasc Pharmacol Ther, 3(4), 319-326.

Kashima, K., Yokoyama, S., Daa, T., Nakayama, I., & Iwaki, T. (1997). Immunohistochemical study on tissue transglutaminase and copper-zinc superoxide dismutase in human myocardium: its relevance to apoptosis detected by the nick end labelling method. Virchows Arch, 430(4), 333-338.

Kikuchi, R., Tsuda, H., Kanai, Y., Kasamatsu, T., Sengoku, K., Hirohashi, S., Inazawa, J., & Imoto, I. (2007). Promoter hypermethylation contributes to frequent inactivation of a putative conditional tumor suppressor gene connective tissue growth factor in ovarian cancer. Cancer Res, 67(15), 7095-7105.

Kim, J. H., Choy, H. E., Nam, K. H., & Park, S. C. (2001).. 928(1), 65-70.

Klingberg, F., Hinz, B., & White, E. S. (2013). The myofibroblast matrix: implications for tissue repair and fibrosis. J Pathol, 229(2), 298-309.

McCarron, A., Parsons, D., & Donnelley, M. (2021). Animal? Am J Pathol, 191(2), 228-242.

Murtaugh, M. P., Arend, W. P., & Davies, P. J. (1984). Induction of tissue transglutaminase in human peripheral blood monocytes. J Exp Med, 159(1), 114-125.

Nardacci, R., Lo Iacono, O., Ciccosanti, F., Falasca, L., Addesso, M., Amendola, A., Antonucci, G., Craxì, A., Fimia, G. M., Iadevaia, V., Melino, G., Ruco, L., Tocci, G., Ippolito, G., & Piacentini, M. (2003). Transglutaminase Type II Plays a Protective Role in Hepatic Injury. The American Journal of Pathology, 162(4), 1293-1303.

Park, S. C., Yeo, E. J., Han, J. A., Hwang, Y. C., Choi, J. Y., Park, J. S., Park, Y. H., Kim, K. O., Kim, I. G., Seong, S. C., & Kwak, S. J. (1999). Aging process is accompanied by increase of transglutaminase C. J Gerontol A Biol Sci Med Sci, 54(2), B78-83.

Penumatsa, K. C., Toksoz, D., Warburton, R. R., Kharnaf, M., Preston, I. R., Kapur, N. K., Khosla, C., Hill, N. S., & Fanburg, B. L. (2017). Transglutaminase 2 in pulmonary and cardiac tissue remodeling in experimental pulmonary hypertension. 313(5), L752-L762.

Petrak, J., Pospisilova, J., Sedinova, M., Jedelsky, P., Lorkova, L., Vit, O., Kolar, M., Strnad, H., Benes, J., Sedmera, D., Cervenka, L., & Melenovsky, V. (2011). Proteomic and transcriptomic analysis of heart failure due to volume overload in a rat aorto-caval fistula model provides support for new potential therapeutic targets - monoamine oxidase A and transglutaminase 2. Proteome Science, 9(1), 69-69.

Piacentini, M., Farrace, M. G., Hassan, C., Serafini, B., & Autuori, F. (1999). 'Tissue' transglutaminase release from apoptotic cells into extracellular matrix during human liver fibrogenesis. J Pathol, 189(1), 92-98.

Popov, Y., Sverdlov, D. Y., Sharma, A. K., Bhaskar, K. R., Li, S., Freitag, T. L., Lee, J., Dieterich, W., Melino, G., & Schuppan, D. (2011). Tissue Transglutaminase Does Not Affect Fibrotic Matrix Stability or Regression of Liver Fibrosis in Mice. Gastroenterology, 140(5), 1642-1652.

Sajid, M. I., Moazzam, M., Kato, S., Yeseom Cho, K., & Tiwari, R. K. (2020). Overcoming Barriers for siRNA Therapeutics: From Bench to Bedside. Pharmaceuticals (Basel), 13(10).

Sarang, Z., Molnár, P., Németh, T., Gomba, S., Kardon, T., Melino, G., Cotecchia, S., Fésüs, L., & Szondy, Z. (2005). Tissue transglutaminase (TG2) acting as G protein protects hepatocytes against Fas-mediated cell death in mice. Hepatology, 42(3), 578-587.

Saxena, S., Jónsson, Z. O., & Dutta, A. (2003). Small RNAs with imperfect match to endogenous mRNA repress translation. Implications for off-target activity of small inhibitory RNA in mammalian cells. J Biol Chem, 278(45), 44312-44319.

Scacheri, P. C., Rozenblatt-Rosen, O., Caplen, N. J., Wolfsberg, T. G., Umayam, L., Lee, J. C., Hughes, C. M., Shanmugam, K. S., Bhattacharjee, A., Meyerson, M., & Collins, F. S. (2004). Short interfering RNAs can induce unexpected and divergent changes in the levels of untargeted proteins in mammalian cells. Proc Natl Acad Sci U S A, 101(7), 1892-1897.

Shinde, A. V., Dobaczewski, M., de Haan, J. J., Saxena, A., Lee, K. K., Xia, Y., Chen, W., Su, Y., Hanif, W., Kaur Madahar, I., Paulino, V. M., Melino, G., & Frangogiannis, N. G. (2017). Tissue transglutaminase induction in the pressure-overloaded myocardium regulates matrix remodelling. Cardiovasc Res, 113(8), 892-905.

Shweke, N., Boulos, N., Jouanneau, C., Vandermeersch, S., Melino, G., Dussaule, J.-C., Chatziantoniou, C., Ronco, P., & Boffa, J.-J. (2008). Tissue Transglutaminase Contributes to Interstitial Renal Fibrosis by Favoring Accumulation of Fibrillar Collagen through TGF-β Activation and Cell Infiltration. The American Journal of Pathology, 173(3), 631-642.

Small, K., Feng, J. F., Lorenz, J., Donnelly, E. T., Yu, A., Im, M. J., Dorn, G. W., 2nd, & Liggett, S. B. (1999). Cardiac specific overexpression of transglutaminase II (G(h)) results in a unique hypertrophy phenotype independent of phospholipase C activation. J Biol Chem, 274(30), 21291-21296.

Song, H., Kim, B. K., Chang, W., Lim, S., Song, B. W., Cha, M. J., Jang, Y., & Hwang, K. C. (2011). Tissue transglutaminase 2 promotes apoptosis of rat neonatal cardiomyocytes under oxidative stress. J Recept Signal Transduct Res, 31(1), 66-74.

Stephens, P., Grenard, P., Aeschlimann, P., Langley, M., Blain, E., Errington, R., Kipling, D., Thomas, D., & Aeschlimann, D. (2004). Crosslinking and G-protein functions of transglutaminase 2 contribute differentially to fibroblast wound healing responses. J Cell Sci, 117(Pt 15), 3389-3403.

Steppan, J., Sikka, G., Jandu, S., Barodka, V., Halushka, M. K., Flavahan, N. A., Belkin, A. M., Nyhan, D., Butlin, M., Avolio, A., Berkowitz, D. E., & Santhanam, L. (2014). Exercise. 3(2), e000599.

Szondy, Z., Korponay-Szabó, I., Király, R., Sarang, Z., & Tsay, G. J. (2017). Transglutaminase 2 in human diseases. BioMedicine, 7(3), 15.

Szondy, Z., Mastroberardino, P. G., Váradi, J., Farrace, M. G., Nagy, N., Bak, I., Viti, I., Wieckowski, M. R., Melino, G., Rizzuto, R., Tósaki, Á., Fesus, L., & Piacentini, M. (2006). Tissue transglutaminase (TG2) protects cardiomyocytes against ischemia/reperfusion injury by regulating ATP synthesis. Cell Death & Differentiation, 13(10), 1827-1829.

Telci, D., & Griffin, M. (2006). Tissue transglutaminase (TG2)--a wound response enzyme. Front Biosci, 11, 867-882.

Thacher, S. M., & Rice, R. H. (1985). Keratinocyte-specific transglutaminase of cultured human epidermal cells: relation to cross-linked envelope formation and terminal differentiation. Cell, 40(3), 685-695.

Tuggle, K. L., Birket, S. E., Cui, X., Hong, J., Warren, J., Reid, L., Chambers, A., Ji, D., Gamber, K., Chu, K. K., Tearney, G., Tang, L. P., Fortenberry, J. A., Du, M., Cadillac, J. M., Bedwell, D. M., Rowe, S. M., Sorscher, E. J., & Fanucchi, M. V. (2014). Characterization of defects in ion transport and tissue development in cystic fibrosis transmembrane conductance regulator (CFTR)-knockout rats. PLOS ONE, 9(3), e91253.

Verderio, E., & Griffin, M. (1999). Cell-surface tissue transglutaminase regulates matrix storage of latent TGF-beta binding protein-1 (LTBP-1) and fibronectin accumulation. In).

Wang, Z., Stuckey, D. J., Murdoch, C. E., Camelliti, P., Lip, G. Y. H., & Griffin, M. (2018). Cardiac fibrosis can be attenuated by blocking the activity of transglutaminase 2 using a selective small-molecule inhibitor. Cell death & disease, 9(6), 613-613.

Wang, Z., Telci, D., & Griffin, M. (2011). Importance of syndecan-4 and syndecan -2 in osteoblast cell adhesion and survival mediated by a tissue transglutaminase-fibronectin complex. Exp Cell Res, 317(3), 367-381.

Wess, L., Eastwood, M. A., Wess, T. J., Busuttil, A., & Miller, A. (1995). Cross linking of collagen is increased in colonic diverticulosis. Gut, 37(1), 91-94.

Xu, D., Xu, N., Sun, L., Yang, Z., He, M., & Li, Y. (2022). TG2 as a novel breast cancer prognostic marker promotes cell proliferation and glycolysis by activating the MEK/ERK/LDH pathway. BMC Cancer, 22(1), 1267.

Yang, M. H., Lin, B. R., Chang, C. H., Chen, S. T., Lin, S. K., Kuo, M. Y. P., Jeng, Y. M., Kuo, M. L., & Chang, C. C. (2013). Erratum: Connective tissue growth factor modulates oral squamous cell carcinoma invasion by activating a miR-504/FOXP1 signalling. Oncogene, 32(5), 670-670.

Zhang, Z., Vezza, R., Plappert, T., McNamara, P., Lawson, J. A., Austin, S., Praticò, D., Sutton, M. S.-J., & FitzGerald, G. A. (2003). 92(10), 1153-1161.

Zhen, Y., Ye, Y., Yu, X., Mai, C., Zhou, Y., Chen, Y., Yang, H., Lyu, X., Song, Y., Wu, Q., Fu, Q., Zhao, M., Hua, S., Wang, H., Liu, Z., Zhang, Y., & Fang, W. (2013). Reduced CTGF Expression Promotes Cell Growth, Migration, and Invasion in Nasopharyngeal Carcinoma. PLOS ONE, 8(6), e64976.

---

## [Editor Report · Decision Letter 1]

20 Jan 2023

Involvement and possible role of transglutaminases 1 and 2 in mediating fibrotic signalling, collagen cross-linking and cell proliferation in neonatal rat ventricular fibroblasts

PONE-D-22-29124R1

Dear Dr. Al-u'datt,

We’re pleased to inform you that your manuscript has been judged scientifically suitable for publication and will be formally accepted for publication once it meets all outstanding technical requirements.

Kind regards,

Michael Bader

Academic Editor

PLOS ONE
---

## [Editor Report · Acceptance letter]

17 Feb 2023

PONE-D-22-29124R1 

Involvement and possible role of transglutaminases 1 and 2 in mediating fibrotic signalling, collagen cross-linking and cell proliferation in neonatal rat ventricular fibroblasts 

Dear Dr. Al-u'datt:

I'm pleased to inform you that your manuscript has been deemed suitable for publication in PLOS ONE. Congratulations! Your manuscript is now with our production department. 

Kind regards, 

on behalf of

Prof. Michael Bader 

Academic Editor

PLOS ONE